# See the Big in the Small: Budget-Friendly Explanations for Large Language Models

## Abstract

With Large language models (LLMs) becoming increasingly prevalent in various applications, the need for interpreting their predictions has become a critical challenge. As LLMs vary in architecture and some are closed-sourced, model-agnostic techniques show great promise without requiring access to the model's internal parameters. However, existing model-agnostic techniques require obtaining an LLM's outputs on a large number of perturbed samples, which leads to high economic costs. To address this limitation, we propose to leverage explanations from budget-friendly models as proxies to explain expensive LLMs, and a corresponding simple yet effective screen-and-apply framework to ensure the faithfulness of applying proxy explanations. We empirically evaluate our approach through a series of empirical studies, demonstrating that proxy explanations can achieve over 90% fidelity compared to oracle explanations, while requiring only 11% of the cost of oracle explanations. Moreover, we show that such proxy explanations also perform well on downstream tasks such as optimizing LLM's performance in in-context learning. Additionally, we open-source our code and datasets to facilitate future research in this area[1].

## 1 Introduction

As large language models (LLMs) become increasingly prevalent across a wide range of applications, the demand for interpreting their predictions to end-users has grown accordingly. Given the rapid evolution and variety of LLM architectures, coupled with the widespread use of closed-source models such as GPT-4o (Achiam et al., 2023), Google Gemini (et al., 2024), and Claude (Anthropic, 2024), model-agnostic explanation techniques have become particularly appealing due to their independence from model internals. To ensure the understandability for end-users, the complexity of LLMs often necessitates local explanations, which describe the model's behavior in the vicinity of a specific input instance (Zhao et al., 2024).

Although existing local model-agnostic explanation methods (Ribeiro et al., 2018; 2016; Lundberg & Lee, 2017; Guidotti et al., 2018) can be adapted to LLMs (Liu & Zhang, 2025; Paes et al., 2024), the economic cost can be substantial for state-of-the-art commercial models. Local model-agnostic methods use perturbation models to create samples from the local neighborhood around the input instance and observe corresponding model outputs. To ensure faithful explanations, these methods must learn explanations on a sufficiently large set of input-output pairs, resulting in substantial charges from invoking the LLM APIs.

Let us take using LIME (Ribeiro et al., 2016) to explain GPT-4o (Achiam et al., 2023) on a simple sentiment analysis task as an example. LIME typically generates approximately 5000 perturbation samples and queries GPT-4o for its predictions on these samples. As detailed in Table 1, GPT-4o charges $12.50 per million tokens for inputs and $10.00 per million tokens for outputs. Assuming an average input length of 1000 tokens, explaining a single input instance would cost around $12.50. This expense may be prohibitive for individual users. Repeated queries can quickly accumulate significant costs, posing a barrier in non-commercial or research contexts.

To address this limitation, we propose to generate proxy explanations for expensive models by sampling from budget-friendly ones. Figure 1 details our idea of using explanations generated from

---

[1]The code and datasets are available at `https://anonymous.4open.science/r/XLLM-Bench`

Figure 1: The workflow for leveraging proxy explanations from budget-friendly models to reduce the cost of explaining expensive LLMs.

budget-friendly models as a proxy explanation for expensive LLMs to reduce the explanation generation cost. Our approach is motivated by a key observation: We observe that different models can exhibit similar behaviors. Especially when they produce the same outputs, they also tend to behave alike on similar inputs. Recent progress has produced many open-source and smaller-scale LLMs (Liu et al., 2024; Yang et al., 2025; Dubey et al., 2024) that are both cost-efficient and highly capable, making them suitable candidates for generating proxy explanations. To ensure reliability, it is essential to verify that if proxies can be faithfully used on a specific task or input. To this end, we introduce a *screen-and-apply* framework: before applying proxy explanations, users can first use a simple screening step to check if using proxy explanations from the budget-friendly model is appropriate for the specific task or input.

We conducted a series of empirical studies among 12 state-of-the-art LLMs on 3 datasets, 7 tasks. These models include both close and open-source LLMs at various scales: GPT-4o and GPT-4o Mini (Achiam et al., 2023), Qwen 2.5 series (0.5B to 72B parameters) (Yang et al., 2025), DeepSeek V3 (Liu et al., 2024), and LLaMA 3.1 series (8B and 70B parameters) (Dubey et al., 2024); the datasets span seven tasks: five representative subjects from the MMLU benchmark (Hendrycks et al., 2020), the sentiment classification dataset SST-2 (Socher et al., 2013), and the Google Natural Questions (NQ) dataset (Kwiatkowski et al., 2019). We generate explanations using two mainstream methods, LIME (Ribeiro et al., 2016) and SHAP (Lundberg & Lee, 2017). The results show that our framework can effectively help users generate faithful proxy explanations while reducing the cost by 88.2%. Moreover, we also show that the proxy explanations are effective in downstream tasks, such as improving the performance of expensive LLMs in few-shot learning.

To make our findings accessible by the research community, we collect and release perturbation samples used in our empirical studies, and we call it XLLM-Bench dataset. This dataset can serve as a foundation for future research related to explaining large language models.

**Contributions.** We make the following contributions:

- To address the high cost of generating local model-agnostic explanations for LLMs, we propose a screen-and-apply proxy-explanation generation framework, which helps users to effectively use budget-friendly models to generate proxy explanations for expensive LLMs.

- Our empirical studies demonstrate that our framework can provide faithful proxy explanations while significantly reducing the cost by 88.2%, and perform well on downstream tasks such as improving the performance of expensive LLMs in few-shot learning.

- We release XLLM-Bench, which is a large-scale dataset of perturbation samples used in our empirical studies, to facilitate further research in explaining LLMs.

## 2 BACKGROUND

In this section, we introduce the background knowledge and notations used in this paper.

## 2.1 Large Language Models

We consider a large language model as a probabilistic function $f$ that maps an input sequence of tokens $\boldsymbol{x} = [x_1, x_2, \ldots, x_t]$ to a probability distribution over the next possible token, denoted as $f(\boldsymbol{x})$. Formally, we define $f : \mathcal{V}^* \to \mathbb{R}^{|\mathcal{V}|}$, where $\mathcal{V}$ is the vocabulary set, and the output is a probability distribution over the vocabulary. As LLMs usually have a fixed maximum input length $n$, we assume $\boldsymbol{x} \in \mathbb{X}^n$.

## 2.2 Local Model-Agnostic Explanation Techniques

A local model-agnostic explanation technique $t$ takes as input a predictive model $f$ and an instance $\boldsymbol{x}$, and returns an explanation $g_{f,\boldsymbol{x}}$ that describes the model's behavior in the local neighborhood of $\boldsymbol{x}$.

In this paper, we primarily focus on attribution-based techniques, as they are the most popular form of local explanations. Attribution-based explanations assign importance scores to the features of an input $\boldsymbol{x} \in \mathbb{X}$ to quantify their contributions to the model's prediction $f(\boldsymbol{x})$. The resulting explanation $g_{f,\boldsymbol{x}}$ is expressed as a feature attribution vector $\boldsymbol{a} = [a_1, a_2, \ldots, a_n]$, where each $a_i$ represents the individual contribution of the corresponding feature $x_i$ to the prediction. Formally, an attribution-based local explanation technique $t$ can be defined as a function:

$$t : \mathbb{F} \times \mathbb{X}^n \to \mathbb{R}^n,$$

where $\mathbb{F}$ is the space of predictive models, $\mathbb{X}^n$ is the input space, and $n$ is the dimensionality of the input $\boldsymbol{x}$.

## 2.3 Desiderata of Explanations

**Fidelity** and **understandability** are two key desiderata for local explanations aimed at end-users (Dwivedi et al., 2023; Zhang et al., 2021; Rojat et al., 2021; Mahto, 2025).

On one hand, explanations should faithfully reflect the model's decision-making process. High fidelity indicates that the explanation accurately captures how the model arrives at its predictions. On the other hand, explanations should be understandable—that is, they should be presented in a form that humans can easily interpret.

In this paper, we conduct our empirical studies using attribution-based techniques, which produce simple and understandable forms of explanation. Therefore, we focus our evaluation on the **fidelity** of proxy explanations generated by budget-friendly models.

## 3 Proxy Explanation Framework

In this section, we introduce the proxy explanations framework, which consists of two main steps: (1) a screening step to determine if proxy explanations can be reliably used for current tasks or instances, and (2) applying the proxy explanations to generate faithful explanations for the target expensive LLMs.

### 3.1 Screening Step

We use a two-stage screening to ensure proxy explanations from a budget-friendly model $f'$ are reliable for an expensive LLM $f$ on a task or dataset with input set $\mathbb{D}$ using a local technique $t$. Specifically, the screening procedure includes: an offline task-level screening and an online instance-level screening. 1) The task-level screening is performed once per task. It assesses whether $f'$ can provide sufficiently faithful proxy explanations for $f$ over the entire input set $\mathbb{D}$, offering a task-level fidelity assessment. 2) The instance-level screening is a lightweight runtime check applied to each input $\boldsymbol{x}$. It verifies whether $f'$ and $f$ agree on the prediction for $\boldsymbol{x}$.

**Task-Level Screening (Offline)** Given a target LLM $f$, a dataset or task with input set $\mathbb{D}$, and an explanation technique $t$, we run task-level screening once to ensure that proxy explanations from a budget-friendly model $f'$ are on average sufficiently faithful. Specifically, we perform statistical

hypothesis testing to check whether the proxy explanations from $f'$ achieve at least a fraction $\tau$ of the fidelity of oracle explanations from $f$, with confidence level $1-\delta$. Formally, we define the task-level screening decision as a binary function: $s_{\text{task}}^{\tau,\delta}(\mathbb{D}; f, f') \in \{0, 1\}$. To keep consistent with the instance-level screening, we only consider inputs on which $f$ and $f'$ agree: $\mathbb{D}' = \{\boldsymbol{x} \in \mathbb{D} \ : \ f(\boldsymbol{x}) = f'(\boldsymbol{x})\}$, from which we draw samples via rejection sampling. Let $q_{\text{proxy}}(\boldsymbol{x})$ and $q_{\text{oracle}}(\boldsymbol{x})$ denote the (per-instance) fidelities on $\mathbb{D}'$ of proxy and oracle explanations, respectively (as defined in Section 4.2). We conduct a *sequential one-sided paired $t$-test* on the paired differences

$$d_i \ = \ q_{\text{proxy}}(\boldsymbol{x}_i) \ - \ \tau \, q_{\text{oracle}}(\boldsymbol{x}_i), \qquad i = 1, \dots, n,$$

and test

$$H_0 : \ \mu_d < 0 \quad \text{vs.} \quad H_1 : \ \mu_d \geq 0,$$

where $\mu_d = \mathbb{E}[d_i]$ is the population mean difference on $\mathbb{D}'$. At step $n$ we update the sample mean and variance of the paired differences,

$$\bar{d} \ = \ \frac{1}{n} \sum_{i=1}^{n} d_i, \qquad s_d^2 \ = \ \frac{1}{n-1} \sum_{i=1}^{n} (d_i - \bar{d})^2.$$

After each new paired sample, we compute a $(1-\delta)$ confidence interval for $\mu_d = \bar{q}_{\text{proxy}} - \tau \, \bar{q}_{\text{oracle}}$ as

$$\left( \bar{d} - t_{\nu, \, 1-\delta/2} \, \frac{s_d}{\sqrt{n}} \ , \ \ \bar{d} + t_{\nu, \, 1-\delta/2} \, \frac{s_d}{\sqrt{n}} \right),$$

where $t_{\nu, \, 1-\delta/2}$ is the $1 - \delta/2$ quantile of the $t$-distribution with $\nu = n - 1$ degrees of freedom.

If the entire interval lies above zero, we accept $H_1$; if it lies entirely below zero, we accept $H_0$. Otherwise, we continue sampling until a confident decision is reached or a maximum sample size $N$ is exhausted. Finally, if $H_1$ is accepted, we set $s_{\text{task}}^{\tau,\delta}(f'; f, \mathbb{D}) = 1$, indicating that proxy explanations from $f'$ are sufficiently faithful on average for $\mathbb{D}$; otherwise, we set $s_{\text{task}}^{\tau,\delta}(f'; f, \mathbb{D}) = 0$.

**Instance-Level Screening (Online)** If $f'$ passes the task-level screening, we apply an instance-level check for each input $\boldsymbol{x}$ to filter out cases where the two models disagree. For a given $\boldsymbol{x}$, the instance-level screening function is

$$s_{\text{inst}}(\boldsymbol{x}; f, f') \ = \ \mathbf{1}[\, f(\boldsymbol{x}) = f'(\boldsymbol{x}) \,].$$

The rationale is twofold: (1) local explanations are designed for the model's current prediction, so proxy explanations are appropriate only when the two models agree; and (2) disagreement suggests different local decision behavior around $\boldsymbol{x}$, making proxy explanations more likely to be unfaithful.

## 3.2 Applying Proxy Explanations

If the budget-friendly model $f'$ passes the task-level screening, and the input instance $\boldsymbol{x}$ passes the instance-level screening, our framework will apply the proxy explanations from $f'$, i.e., $t(f', \boldsymbol{x})$, to explain the behavior of the expensive LLM $f$ around $\boldsymbol{x}$.

For more details, please refer to Appendix D.

## 4 Fidelity Evaluation

In this section, we first introduce the experimental setup used in our empirical studies, and then present and analyze results to answer the following research questions:

1. **Cost Reduction:** To what extent can the proposed proxy explanation framework reduce the cost of generating explanations for expensive LLMs? This is the primary focus of our study.
2. **Screening Reliability:** How reliable is the screening step in our framework? This checks if the screening step is necessary and sufficient to ensure the fidelity of proxy explanations.
3. **Proxy Explanation Generalizability:** Does the transferability of explanations across models hold consistently across different tasks and datasets? This aspect is crucial for demonstrating the generalizability and applicability of our method.

Table 1: Common LLM Official API pricing (USD per million tokens). Specifically, Qwen 2.5 models with 0.5B and 1.5B parameters can be accessed from Alibaba (https://www.aliyun.com/) for **free**, and all open-source models with 8B or fewer parameters can be run locally on a single consumer-grade GPU.

| Model name | GPT-4o | | DeepSeek V3 | Qwen 2.5 | | | | | | | LLaMA 3.1 | |
|---|---|---|---|---|---|---|---|---|---|---|---|---|
| Model size | Regular | Mini | 685B | 0.5B | 1.5B | 3B | 7B | 14B | 32B | 72B | 8B | 70B |
| **Input** | $2.50 | $0.15 | $0.27 | – | – | $0.04 | $0.07 | $0.14 | $0.28 | $0.56 | $0.18 | $0.88 |
| **Output** | $10.00 | $0.60 | $1.10 | – | – | $0.12 | $0.14 | $0.41 | $0.83 | $1.67 | $0.18 | $0.88 |
| **Open-source** | No | No | Yes | Yes | Yes | Yes | Yes | Yes | Yes | Yes | Yes | Yes |

## 4.1 EXPERIMENTAL SETUP

### 4.1.1 TARGET MODELS AND EXPLANATION TECHNIQUES

We conducted our experiments on 12 popular generative language models, including two from the GPT-4o series, DeepSeek V3, seven Qwen 2.5 models, and two Llama 3.1 models, as listed in Table 1. We accessed GPT-4o series and DeepSeek V3 via their official APIs, while running the other models locally.

The models were selected based on their popularity, as well as diversity in architecture, size, and pricing. They cover both dense and Mixture-of-Experts (MoE) architectures, parameter counts ranging from 0.5 billion (Qwen2.5-0.5B) to 685 billion (DeepSeek V3). Their associated costs vary significantly: GPT-4o is the most expensive at $2.50 per million input tokens and $10.00 per million output tokens, while Qwen2.5-0.5B is the most affordable, whose official APIs are currently free and can also be deployed locally on a single consumer-grade GPU with minimal computational costs.

We use two representative attribution-based explanation techniques: LIME (Ribeiro et al., 2016) and Kernel SHAP (Lundberg & Lee, 2017), to generate local explanations. For both methods, we set the number of perturbation samples to 1,000 and use default values for all other hyperparameters. When applying our proxy explanation framework, we set the dataset-level screening threshold $\tau = 0.9$ and confidence level $1 - \delta = 0.99$, and a maximum sample size $N = 50$. We also conducted sensitivity analysis on hyperparameters in Appendix F.

### 4.1.2 TASKS AND DATASETS

We evaluate our approach on three representative tasks: sentiment analysis, multiple-choice question answering, and text generation, where sentiment analysis is a classic task in studying model explanations, multiple-choice question answering is a common benchmark for evaluating the performance of LLMs, and text generation is a widely used task of LLMs beyond classification. Considering the budget limitation, we select one dataset from each task to analyze the effectiveness of using proxy explanations. Besides, we conduct our experiments on another three datasets to further validate whether the croess-model explanation transferability holds across different datasets.

**Sentiment Analysis** We use the Stanford Sentiment Treebank (SST) dataset (Socher et al., 2013) for classification. Following the standard train/validation/test split, we generate explanations for all 2,210 sentences in the test set. The target model is prompted in a zero-shot setting to predict whether the sentiment of a given sentence is positive or negative.

**Multiple-Choice Question Answering** We use the MMLU dataset (Hendrycks et al., 2020), which contains 57,000 questions spanning 57 topics. We select 5 topics for evaluation: high school chemistry, high school physics, microeconomics, world history, and computer science. For each topic, we use the questions in the validation set as the in context examples and the questions in the test set as the target questions. We generate explanations for all 1321 questions in the test set.

**Text Generation** We use the Google Natural Questions (NQ) dataset (Kwiatkowski et al., 2019) for text generation. We randomly select 200 questions from the validation set and generate short answers using the target models. To apply LIME and Kernel SHAP, we follow prior work (Luss et al., 2024; Hackmann et al., 2024; Liu & Zhang, 2025; Paes et al., 2024) in using a scoring function $f_s : \mathcal{X} \to \mathbb{R}$ that maps the generated sequence to a scalar score, effectively framing the text generation task as a regression problem. In our experiments, we use `all-MiniLM-L6-v2` (Wang et al., 2020) from

Table 2: Cost Reduction Ratios (CRR) achieved by using the proxy explanation framework to explain expensive LLMs with LIME and Kernel SHAP. Here, $CRR_{mean}$ and $CRR_{max}$ denote the average and maximum CRR obtained from screened budget-friendly models with API access. $CRR_{local}$ also denotes the maximum CRR, but we run all budget-friendly models locally, thus further reducing the cost.

| | | LIME | | | Kernel SHAP | | |
|---|---|---|---|---|---|---|---|
| **Target Model** | | SST | MMLU | NQ | SST | MMLU | NQ |
| GPT-4o | $CRR_{mean}$ | 8.74 | 3.41 | 5.70 | 8.90 | 3.41 | 7.29 |
| | $CRR_{max}$ | 10.33 | 4.84 | 7.41 | 10.33 | 4.84 | 8.20 |
| | $CRR_{local}$ | 14.17 | 5.62 | 10.53 | 14.17 | 5.62 | 11.11 |
| GPT-4o mini | $CRR_{mean}$ | 2.50 | 1.83 | 3.65 | 1.97 | 1.96 | 3.43 |
| | $CRR_{max}$ | 3.08 | 2.88 | 6.67 | 3.10 | 3.19 | 6.67 |
| | $CRR_{local}$ | 13.15 | 4.98 | 6.67 | 14.44 | 5.78 | 10.53 |
| DeepSeek V3 | $CRR_{mean}$ | 3.06 | 2.15 | 2.40 | 3.88 | 2.27 | 3.85 |
| | $CRR_{max}$ | 4.60 | 3.05 | 4.17 | 6.33 | 3.16 | 7.41 |
| | $CRR_{local}$ | 13.31 | 5.32 | 8.33 | 13.64 | 6.10 | 8.33 |
| Qwen 2.5 14B | $CRR_{mean}$ | 2.39 | 1.82 | 2.72 | 1.85 | 1.90 | 3.49 |
| | $CRR_{max}$ | 2.90 | 2.99 | 6.06 | 2.89 | 3.20 | 6.25 |
| | $CRR_{local}$ | 17.13 | 5.64 | 11.76 | 17.13 | 6.10 | 11.76 |
| Qwen 2.5 32B | $CRR_{mean}$ | 3.18 | 2.06 | 3.74 | 3.19 | 2.16 | 3.86 |
| | $CRR_{max}$ | 4.77 | 3.15 | 6.06 | 4.77 | 3.05 | 6.06 |
| | $CRR_{local}$ | 15.24 | 5.30 | 9.09 | 15.24 | 5.31 | 9.09 |
| Qwen 2.5 72B | $CRR_{mean}$ | 5.10 | 2.47 | 3.97 | 5.05 | 2.76 | 5.13 |
| | $CRR_{max}$ | 7.04 | 3.40 | 6.25 | 6.91 | 3.66 | 7.14 |
| | $CRR_{local}$ | 16.25 | 6.07 | 9.09 | 16.25 | 6.31 | 10.53 |
| Llama 3.1 70B | $CRR_{mean}$ | 5.32 | 2.92 | 3.72 | 6.14 | 2.93 | 4.92 |
| | $CRR_{max}$ | 6.74 | 3.96 | 5.13 | 8.02 | 3.96 | 6.67 |
| | $CRR_{local}$ | 10.33 | 5.77 | 6.90 | 17.13 | 5.77 | 10.00 |

the Sentence-Transformers library (Reimers & Gurevych, 2019) as the scoring function. This pretrained sentence transformer encodes each generated answer into a semantic embedding vector, and we use the cosine similarity between the sample outputs and target outputs as the final scalar score.

## 4.2 Fidelity Metrics

LIME and Kernel SHAP construct a local surrogate model to approximate the target model's behavior. Following Balagopalan et al. (2022); Yeh et al. (2019); Ismail et al. (2021), given a target model $f$, an input $\boldsymbol{x}$, a surrogate explanation model $g$, a neighborhood distribution $D(\boldsymbol{x})$, and a performance metric $L$ (e.g., accuracy, AUROC, or mean squared error (MSE)), the (in)fidelity is defined as: $\mathbb{E}_{\boldsymbol{z} \sim D(\boldsymbol{x})} L(f(\boldsymbol{z}), g(\boldsymbol{z}))$. In our experiments, we use **accuracy** as the performance metric $L$.

## 4.3 Evaluation Results

### 4.3.1 RQ1: Cost Reduction of Explaining Expensive LLMs

We use Cost Reduction Ratio (CRR) to measure the cost reduction achieved by our proxy explanation framework compared to directly generating explanations from expensive LLMs. Specifically, when explaining an expensive LLM $f$ with a budget-friendly model $f'$ on input set $\mathbb{D}$ that passes the task-level screening step, we define the CRR as:

$$\frac{\sum_{x \in \mathbb{D}} \text{Cost}(f, x)}{\sum_{x \in \mathbb{D}} \text{Cost}(f', x) \cdot s_{\text{inst}}(\boldsymbol{x}; f, f') + \sum_{x \in \mathbb{D}} \text{Cost}(f, x) \cdot (1 - s_{\text{inst}}(\boldsymbol{x}; f, f')) + \text{Cost}_{\text{screen}}(f, f', \mathbb{D})}$$

where $\text{Cost}(f, \boldsymbol{x})$ denotes the cost of generating explanations for model $f$, $s(\boldsymbol{x}; f, f')$ is the instance-level screening function defined in Section 3, and $\text{Cost}_{\text{screen}}(f, f', \mathbb{D})$ is the cost of performing task-level screening on dataset $\mathbb{D}$. Here, we split the models into two groups based on their costs: all models that can be run locally on a single consumer-grade GPU are considered budget-friendly models, while the rest are classified as target expensive models.

Table 2 shows the CRR achieved by using our proxy explanation framework to explain expensive LLMs with LIME and Kernel SHAP. We can see that for each expensive model, the use of a budget-friendly proxy model significantly reduces the cost of generating explanations. Especially for the

Table 3: Screening recall, precision, and F1-score of the Proxy Explanation Framework.

| Method | LIME | | | Kernel SHAP | | |
|---|---|---|---|---|---|---|
| Datasets | SST | MMLU | NQ | SST | MMLU | NQ |
| Precision (%) | 100.0 | 99.4 | 94.1 | 100.0 | 100.0 | 100.0 |
| Recall (%) | 80.2 | 77.6 | 76.1 | 96.3 | 97.2 | 96.2 |
| F1-score (%) | 89.0 | 87.2 | 84.2 | 98.1 | 98.5 | 98.0 |

Table 4: Accuracy of proxy LIME explanations on SST, MMLU, and Natural Questions datasets. Budget-friendly models are on the **left**, while target models are on the **top**.

| Proxy \ Target | SST | | | MMLU | | | Natural Questions | | |
|---|---|---|---|---|---|---|---|---|---|
| | Q7B | Q14B | GPT-4o | Q7B | Q14B | GPT-4o | Q7B | Q14B | GPT-4o |
| Q7B | 91.3 | 88.2 | 89.3 | 89.7 | 82.1 | 80.8 | 96.4 | 92.7 | 89.1 |
| Q14B | 86.8 | 92.2 | 86.3 | 81.1 | 89.6 | 82.2 | 95.3 | 97.5 | 90.9 |
| GPT-4o | 88.8 | 87.5 | 91.7 | 79.3 | 81.0 | 88.0 | 97.1 | 96.4 | 97.7 |

most expensive model GPT-4o, using proxy explanations from budget-friendly models can save at most 88% of the cost across all these tasks.

### 4.3.2   RQ2: RELIABILITY OF SCREENING STEP

Our task-level screening verifies whether proxy explanations from a budget-friendly model $f'$ can, on the input set $\mathbb{D}$, achieve fidelity comparable to the oracle explanations generated directly from $f$. For each task, we validate the reliability of the screening step by checking if the screening results align with the actual fidelity of proxy explanations. Since the screening decision is a binary classification problem, we assess its reliability using standard classification metrics: Precision, Recall, and F1-score.

Table 3 shows that our task-level screening step achieves 98.9% precision on average, which means that the screening is sufficient to ensure the fidelity of proxy explanations. For the rare false positives, the realized proxy fidelity still exceeds 89% of the oracle on average, suggesting that even misclassifications remain reasonably faithful. On the other hand, although recall is lower, the results of RQ1 demonstrate that for each expensive model there exists at least one budget-friendly model that passes the screening. Given that budget-friendly models are inexpensive to run, users can screen multiple candidates in parallel to reliably identify a suitable proxy model.

### 4.3.3   RQ3: GENERALIZABILITY OF PROXY EXPLANATIONS

To validate if the cross model explanation transferability holds across different tasks and datasets, we conduct experiments on the datasets described in Section 4.1 and three additional datasets: Large Movie Review (Maas et al., 2011), Fake News (Pérez-Rosas et al., 2018), and Web Question (Berant et al., 2013) for text generation. Overall, we observe that the findings from the three main datasets generally hold across these additional datasets, demonstrating the generalizability of our proxy explanation framework. Due to space limitations, we only show the results of cross-model proxy explanation fidelity between GPT-4o, Qwen2.5-7B, and Qwen2.5-14B on the six tasks in Table 4 and 5. For GPT-4o, Qwen 7B and 14B can both achieve over 90% fidelity compared to the oracle explanations.

For more detailed results and analysis, please refer to Appendix G.

## 5   CASE STUDY

In this section, we present a case study to illustrate how we can leverage proxy explanations in the context of in-context learning (ICL) tasks, which is a core capability of large language models (LLMs) that enables them to perform tasks by conditioning on examples provided in the input prompt, without requiring any parameter updates. Our experiment aims to assess the effectiveness of proxy explanations in guiding ICL prompt optimization. Specifically, we conducted experiments in two contexts: (1) **Prompt Compression** and (2) **Poisoned Examples Removal**.

Table 5: Accuracy of proxy LIME explanations on Large Movie Review, Fake News, and Web Question datasets. Budget-friendly models are on the **left**, while target models are on the **top**.

| Proxy \ Target | Large Movie Review | | | Fake News | | | Web Question | | |
|---|---|---|---|---|---|---|---|---|---|
| | Q7B | Q14B | GPT-4o | Q7B | Q14B | GPT-4o | Q7B | Q14B | GPT-4o |
| Q7B | 75.1 | 72.1 | 70.9 | 83.3 | 79.9 | 78.3 | 91.5 | 84.2 | 84.3 |
| Q14B | 71.4 | 76.3 | 73.5 | 79.1 | 84.5 | 80.2 | 84.3 | 89.1 | 85.2 |
| GPT-4o | 69.5 | 72.0 | 78.4 | 78.4 | 76.3 | 84.0 | 83.9 | 85.0 | 90.1 |

Table 6: Comparison of compression ratios (%) using oracle explanations, proxy explanations, and random deletions. The values represent the average compression ratios across all subjects.

| **Task** | Chemistry | Computer Science | Microeconomics | Psychology | World History |
|---|---|---|---|---|---|
| Oracle Exp. | 49.2 | 50.2 | 72.8 | 72.8 | 57.0 |
| Proxy Exp. | 41.0 | 43.0 | 67.6 | 69.8 | 52.0 |
| Random | 29.0 | 35.6 | 59.8 | 61.0 | 43.4 |

## 5.1 TASK 1: PROMPT COMPRESSION

Prompt compression aims to help users to save costs by reducing the number of examples in the prompt while maintaining the model's performance.

**Experiment Setup** We use explanations to compress the ICL examples in using Qwen 2.5 72B to answer questions from the same five subjects as in Section 4 in the MMLU benchmark (Hendrycks et al., 2020). We compress the prompt by removing the least important examples based on KSHAP explanations. Specifically, we define the prompt compression task as follows: Given a set of examples $S$, an explanation $g$ that attributes the importance of each example, we remove the least important examples based on $g$. The goal is to remove as many examples as possible while ensuring that GPT-4o's performance keep above a certain threshold $\tau$ (we set $\tau = 0.9$). We use the compression ratio as the metric, which is defined as $\text{CompressionRatio}(g) = 1 - \frac{|S_g^\tau|}{|S|}$, where $S_g^\tau$ is the set of examples retained after applying the explanation $g$ and ensuring the model's performance remains above the threshold $\tau$. A higher compression ratio indicates a more effective explanation. We verify if the proxy explanations from a budget-friendly model (Qwen 2.5 7B) can achieve similar performance as oracle explanations from Qwen 2.5 72B. Additionally, we also compare the performance of proxy explanations with a random deletion baseline.

For each subject, we repeat the experiment 15 times with different ICL examples and test questions.

**Evaluation Results** Table 6 shows the results. Proxy explanations generated by Qwen 2.5 7B achieve performance comparable to the oracle explanations from GPT-4o, reaching an average of 91.30% of the oracle's performance. Moreover, they outperform random deletions by a relative margin of 20.55%.

## 5.2 TASK 2: POISONED EXAMPLES REMOVAL

ICL is useful, while poisoning examples can lead to suboptimal performance (Ranjan et al., 2023). Outlier removal focuses on identifying and removing examples that may negatively impact the model's performance, thereby improving the overall quality of the prompt.

**Experiment Setup** We set using GPT-4o to perform sentiment analysis on the SST-2 dataset (Socher et al., 2013) as our target task. We use ICLPoison He et al. (2025) to add poisoning examples to the original dataset, and then use explanations to identify and remove these outliers. We follow the explanation to remove all negatively attributed examples, and evaluate the model's performance after removal. We compare the performance using oracle explanations from GPT-4o and proxy explanations from Qwen 2.5 7B, along with a random deletion baseline.

| Methods | GPT-4o explanation | Qwen 2.5 explanation | Random deletion |
|---|---|---|---|
| Accuracy (%) | 94.2 | 94.0 | 87.1 |

Table 7: Comparison of accuracy (%) using GPT-4o to predict SST-2 sentiment analysis task after using different methods to remove poisoned examples.

**Evaluation Results**   Table 7 shows the results. ICLPoison reduces the accuracy of GPT-4o from 95.1% to 81.5%. Proxy explanations generated by Qwen 2.5 7B achieve performance nearly the same as the oracle explanations from GPT-4o, which recover the accuracy to 94.0% and 94.2% respectively.

## 6   RELATED WORKS

Our work relates to two research directions: model-agnostic explanation techniques for LLMs and methods for reducing the cost of generating post-hoc explanations.

When explaining LLMs in a model-agnostic manner, many approaches (Paes et al., 2024; Enouen et al., 2024; Luss et al., 2024; Hackmann et al., 2024) apply popular explanation methods such as LIME (Ribeiro et al., 2016), SHAP (Lundberg & Lee, 2017), or their variants. These post-hoc techniques typically require multiple queries to the target model and are therefore computationally expensive. An alternative line of work uses LLMs themselves to produce self-explanations, either through chain-of-thought prompting (Wei et al., 2022) or by prompting the model to generate explanations after producing an answer (Ji et al., 2025; Camburu et al., 2018). However, due to the known hallucination issues in LLMs, these generated explanations often lack faithfulness (Parcalabescu & Frank, 2024; Agarwal et al., 2024; Turpin et al., 2023; Madsen et al., 2024).

Several studies have also focused on reducing the cost of generating post-hoc model-agnostic explanations. As surveyed by Chuang et al. (2023a), some methods amortize explanation generation across inputs by training a unified explainer to approximate the distribution of model explanations (Covert et al., 2024; Jethani et al., 2022; Chuang et al., 2023b; Chen et al., 2018a). Other approaches remain non-amortized and generate explanations on a per-instance basis, but aim to improve efficiency through various strategies. These include reducing the number of features in the explanation (Chen et al., 2018b; Yoon et al., 2019; Wang et al., 2022; Jullum et al., 2021), optimizing the perturbation process (Mitchell et al., 2022; Dandolo et al., 2023), or leveraging global dataset-level information (Yu et al., 2025). These methods are orthogonal to our approach and can be integrated with it to further reduce the cost of explanation generation.

## 7   CONCLUSION

In this paper, we introduced a screen-and-apply proxy explanation framework that leverages budget-friendly models to generate proxy explanations for LLMs, reducing the cost of local model-agnostic explanations. We demonstrated the effectiveness of our approach through extensive experiments across various tasks, including text classification, multiple-choice question answering, and text generation. The results indicate that our proxy explanations maintain a high level of fidelity compared to oracle explanations while significantly reducing the cost by 88.2%. We also show that our proxy explanations can enhance the performance of expensive LLMs in few-shot learning scenarios.

## ETHICS STATEMENT

## REPRODUCIBILITY STATEMENT

The code and datasets of our experiments are available at https://anonymous.4open. science/r/XLLM-Bench.

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

## A    THE USE OF LARGE LANGUAGE MODELS

We use LLMs to refine and polish human writing, and find related work with DeepResearch. We do not use LLMs to generate the main content or ideas of this paper.

## B    DATASETS

To reduce the cost of future research on black-box explanation generation for LLMs, improve accessibility, and facilitate reproducibility, we have open-sourced the datasets used in our experiments. In particular, since querying LLMs for perturbed samples is the most computationally expensive part of the process, we provide the model outputs for all perturbed samples used in our experiments. For LIME, perturbations are generated following the original implementation,[2] and for Kernel SHAP, we use the implementation provided by the Captum library.[3] We generate 1000 perturbed samples for each input instance explained.

As mentioned in Section 1, the datasets cover six tasks: five representative subjects from the MMLU benchmark—High School Chemistry, High School Computer Science, High School Microeconomics, High School Psychology, and High School World History—and the SST-2 sentiment classification dataset. For each perturbed sample, we collect the model output logits of the first predicted token.

Additionally, as described in Section 4.1, we select 200 questions from the Natural Questions dataset for the text generation task. We release both the model outputs for each perturbed sample of these questions for reproducibility.

## C    BROADER IMPACTS

This paper proposes using proxy explanations generated from budget-friendly models as substitutes for expensive LLMs, significantly reducing the economic cost of explanation generation. We also release a comprehensive dataset of perturbed samples and corresponding model outputs, which we hope will support and accelerate future research on black-box explanation methods for LLMs. Our approach enables more cost-effective generation of explanations, benefiting both researchers and practitioners, and contributes to improving the explainability, transparency, and fairness of large language models in real-world applications.

However, we acknowledge that while our work focuses on generating local explanations in relatively simple forms, such explanations could potentially be misused—for example, to facilitate adversarial attacks or to form misleading interpretations of model behavior. We encourage responsible use of our framework and dataset, and highlight the importance of developing safeguards when deploying explanation methods in sensitive or high-stakes domains.

## D    PROXY EXPLANATION FRAMEWORK DETAILS

As most budget-friendly models can be run locally, we can perform the task-level screening for multiple budget-friendly models at the same time, as shown in Algorithm 1.

To avoid redundant oracle queries, we introduce a shared buffer that stores each input and the corresponding output from the target model $f$. Formally, we construct

$$\mathcal{B} = \{(\boldsymbol{x}, f(\boldsymbol{x})) : \boldsymbol{x} \in \mathbb{D}\}.$$

This buffer is built once and reused across all candidate proxy models $\{f'_1, \ldots, f'_m\}$.

When each time calculating the fidelity, if the output of $\boldsymbol{x}$ and its neighborhood are already in the buffer $\mathcal{B}$, we can directly use the cached values. This avoids redundant calls to the target model $f$ and speeds up the screening process.

---

[2]https://github.com/marcotcr/lime
[3]https://captum.ai/api/kernel_shap.html

---

**Algorithm 1:** Task-level Screening for Multiple Proxy Models

---

**Input:** Target model $f$; candidate proxy models $\{f'_1, \ldots, f'_m\}$; dataset $\mathbb{D}$; explanation technique $t$; fidelity threshold $\tau$; confidence level $1 - \delta$; maximum sample size $N$.

**Output:** Screening decisions $\{s^{\tau,\delta}_{\text{task}}(f'_j; f, \mathbb{D})\}^m_{j=1}$.

**foreach** *proxy model $f'_j$* **do**

    Initialize $n \leftarrow 0$, paired difference set $\mathcal{D}_j \leftarrow \emptyset$;

    Define $\mathbb{D}'_j = \{\boldsymbol{x} \in \mathbb{D} : f(\boldsymbol{x}) = f'_j(\boldsymbol{x})\}$;

    **while** *decision not reached and $n < N$* **do**

        Sample $\boldsymbol{x}_i$ from $\mathbb{D}'_j$ via rejection sampling;

        Compute fidelities $q_{\text{proxy}}(\boldsymbol{x}_i)$ and $q_{\text{oracle}}(\boldsymbol{x}_i)$ with Buffer $\mathcal{B}$ and update $\mathcal{B}$;

        Form difference $d_i = q_{\text{proxy}}(\boldsymbol{x}_i) - \tau\, q_{\text{oracle}}(\boldsymbol{x}_i)$;

        Update $\hat{d}_j$ and variance $s^2_{d,j}$;

        Construct $(1 - \delta)$ confidence interval for $\mu_d$;

        **if** *interval $> 0$* **then**

            Accept $H_1$; set $s^{\tau,\delta}_{\text{task}}(f'_j; f, \mathbb{D}) = 1$;

        **else**

            **if** *interval $< 0$* **then**

                Accept $H_0$; set $s^{\tau,\delta}_{\text{task}}(f'_j; f, \mathbb{D}) = 0$;

            **else**

                Continue sampling;

    **if** *no decision after $N$ samples* **then**

        Set $s^{\tau,\delta}_{\text{task}}(f'_j; f, \mathbb{D}) = 0$;

---

# E  EXPERIMENTAL SETUP DETAILS

## E.1  MODELS

### E.1.1  DEPLOYMENT

We run Qwen2.5 and Llama3.1 models locally on a machine with total 576 GiB VRAM, while GPT-4o and DeepSeekV3 models are accessed via their official APIs. When locally running the models, we use the default version without additionally quantization or distillation.

## E.2  SENTIMENT ANALYSIS

We perform sentiment analysis using LLMs in a zero-shot setting, where the model is prompted to classify the sentiment of a given sentence as either "positive" or "negative." The sentiment classification task is defined in the system prompt, while the specific sentence to be classified is provided in the user input. The following prompt templates are used:

    **system_prompt**:

```
From now on, you should act as a sentiment analysis
neural network.
You should classify the sentiment of a sentence into
positive or negative.
The input sentence may be empty.  In each task, you
will be given the sentences to be classified, which
end with #####, and then you should reply the
sentiment of the sentence by positive or negative.
```

    **user_prompt**:

**Tactic: Provide examples**

Providing general instructions that apply to all examples is generally more efficient than demonstrating all permutations of a task by example, but in some cases providing examples may be easier. For example, if you intend for the model to copy a particular style of responding to user queries which is difficult to describe explicitly. This is known as "few-shot" prompting.

| | |
|---|---|
| SYSTEM | Answer in a consistent style. |
| USER | Teach me about patience. |
| ASSISTANT | The river that carves the deepest valley flows from a modest spring; the grandest symphony originates from a single note; the most intricate tapestry begins with a solitary thread. |
| USER | Teach me about the ocean. |

Figure 2: The recommended few-shot template provided by OpenAI.

```
Perform the following task, your answer should only be
positive or negative:
Sentence:
{input_sentence}
#####
Sentiment:
```

To obtain the class probabilities, we use the probabilities of the first output token. Specifically, we extract the logits corresponding to the tokens `"positive"` and `"negative"` from the model output and apply the softmax function to compute the probability distribution over the two classes. For local models, we directly obtain the logits and compute the softmax. For GPT-4o and DeepSeekV3, we use their official APIs with the temperature set to 1, retrieve the log probabilities of the target tokens, and then apply the softmax function to obtain normalized probabilities.

### E.2.1 MULTIPLE-CHOICE QUESTION ANSWERING

We perform multiple-choice question answering using LLMs with few-shot prompting, where the model is provided with all examples from the development set using the in-context learning template recommended by OpenAI,[4] as illustrated in Figure 2. We follow the official instructions provided in OpenAI Evals.[5]

The prompt template is as follows:

**System:** `The following are multiple choice questions
(with answers) about {subject}.`
**User:** {example question 1}
**Assistant:** {example answer 1}
⋮
**User:** {question to be answered}

To obtain the class probabilities, we use similar logit extraction methods as in the sentiment analysis task. Specifically, we extract the logits corresponding to the tokens `"A"`, `"B"`, `"C"`, and `"D"` from

---

[4]https://platform.openai.com/docs/guides/prompt-engineering/
six-strategies-for-getting-better-results#tactic-provide-examples
[5]https://github.com/openai/evals/blob/main/examples/mmlu.ipynb

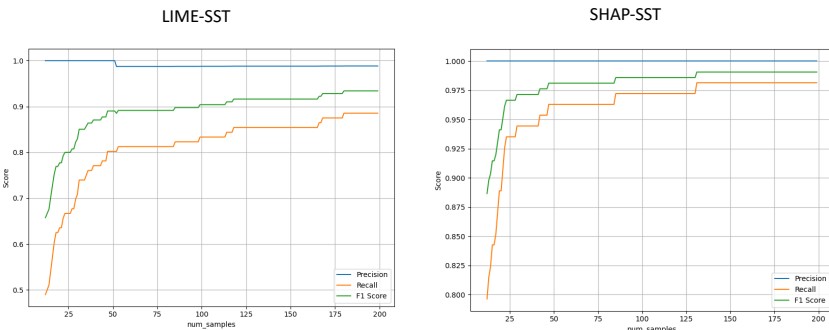

Figure 3: Impact of maximum sample size $N$ on task-level screening results of proxy explanations on the SST dataset.

the model output and apply the softmax function to compute the probability distribution over the four classes.

### E.2.2 TEXT GENERATION

As mentioned in Section 4.1, we treat this task as a regression problem by using a scoring function to evaluate the generated text. When generating text with LLMs, we set the temperature to 1e-5 or set `do_sample = False` to ensure deterministic outputs. We limit the maximum number of generated tokens to 20 and prompt the model to generate short answers, in line with the short-answer format of the Natural Questions dataset. Specifically, we use the following prompt template:

```
[
  {"role": "system", "content":
   "You are a helpful assistant. Answer the question
    briefly, within 10 words. You will be penalized
    for overly long answers."},
  {"role": "user", "content": "{Question}"}
]
```

## F HYPERPARAMETER STUDY

We study the impact of hyperparameters used in our experiments. Since we have chosen commonly used values for $\tau$ and $\delta$ in statistical testing, we primarily investigate the impact of the maximum sample size on the effectiveness of task-level screening.

Figure 3 shows the results of task-level screening on the SST dataset with different maximum sample sizes $N$. We observe that as $N$ increases, the recall of task-level screening also increases, while the precision remains consistently high. As we have demonstrated in Section 4, for each expensive LLM, we can find at least one budget-friendly model that passes the task-level screening, indicating the setting of $N$ does not significantly affect the overall effectiveness of our framework.

## G RQ3: GENERALIZABILITY OF PROXY EXPLANATION DETAILS

In this section, we further analysis the evaluation results.

### G.1 SENTIMENT ANALYSIS

Table 8 and 9 provides the corresponding detailed results, including 95% confidence intervals.

|  | Qwen 0.5B | Qwen 1.5B | Qwen 3B | Qwen 7B | Qwen 14B | Qwen 32B |
|---|---|---|---|---|---|---|
| Qwen 0.5B | **98.81%** ± **0.12** | 41.35% ± 1.19 | 19.26% ± 0.87 | 24.58% ± 1.02 | 18.24% ± 0.86 | 26.35% ± 1.04 |
| Qwen 1.5B | 41.20% ± 1.44 | **86.22%** ± **0.27** | 75.85% ± 0.78 | 79.48% ± 0.67 | 75.41% ± 0.83 | 79.74% ± 0.62 |
| Qwen 3B | 16.72% ± 0.95 | 73.13% ± 0.67 | **91.15%** ± **0.29** | 86.79% ± 0.47 | 89.34% ± 0.42 | 85.23% ± 0.49 |
| Qwen 7B | 23.30% ± 1.15 | 77.50% ± 0.56 | 87.17% ± 0.48 | **91.27%** ± **0.27** | 88.21% ± 0.48 | 87.09% ± 0.41 |
| Qwen 14B | 16.32% ± 0.94 | 72.56% ± 0.71 | 88.37% ± 0.43 | 86.79% ± 0.49 | **92.22%** ± **0.28** | 86.11% ± 0.46 |
| Qwen 32B | 24.57% ± 1.18 | 77.77% ± 0.55 | 86.31% ± 0.51 | 87.72% ± 0.43 | 88.34% ± 0.47 | **90.15%** ± **0.27** |
| Qwen 72B | **26.62%** ± **1.25** | 78.97% ± 0.53 | 85.61% ± 0.54 | 87.89% ± 0.43 | 87.07% ± 0.54 | 88.17% ± 0.36 |
| Llama 8B | 31.55% ± 1.31 | 80.21% ± 0.49 | 82.60% ± 0.63 | 85.92% ± 0.48 | 82.81% ± 0.66 | 84.39% ± 0.50 |
| Llama 70B | 42.88% ± 1.49 | 81.56% ± 0.44 | 74.45% ± 0.86 | 78.97% ± 0.71 | 74.33% ± 0.90 | 79.49% ± 0.67 |
| DeepSeekV3 | 22.65% ± 1.14 | 76.95% ± 0.58 | 87.30% ± 0.48 | 88.28% ± 0.41 | 88.42% ± 0.48 | 87.68% ± 0.39 |
| GPT-4o Mini | 11.87% ± 0.79 | 69.04% ± 0.82 | 88.08% ± 0.45 | 84.63% ± 0.60 | 89.61% ± 0.41 | 83.27% ± 0.59 |
| GPT-4o | 25.52% ± 1.22 | 78.53% ± 0.54 | 86.06% ± 0.53 | 88.79% ± 0.38 | 87.47% ± 0.53 | 87.63% ± 0.38 |

|  | Qwen 72B | Llama 8B | Llama 70B | DeepSeekV3 | GPT-4o Mini | GPT-4o |
|---|---|---|---|---|---|---|
| Qwen 0.5B | 28.12% ± 1.10 | 32.90% ± 1.13 | 42.27% ± 1.26 | 24.45% ± 1.03 | 12.72% ± 0.71 | 26.38% ± 1.10 |
| Qwen 1.5B | 81.09% ± 0.58 | 81.18% ± 0.52 | 83.23% ± 0.44 | 79.19% ± 0.66 | 71.48% ± 0.99 | 80.89% ± 0.62 |
| Qwen 3B | 84.74% ± 0.51 | 80.38% ± 0.60 | 72.51% ± 0.75 | 86.90% ± 0.47 | 90.61% ± 0.45 | 86.05% ± 0.52 |
| Qwen 7B | 87.62% ± 0.39 | 84.15% ± 0.45 | 78.15% ± 0.59 | 88.35% ± 0.40 | 86.90% ± 0.64 | 89.32% ± 0.36 |
| Qwen 14B | 85.12% ± 0.50 | 79.95% ± 0.62 | 72.13% ± 0.78 | 87.09% ± 0.46 | 91.29% ± 0.42 | 86.25% ± 0.52 |
| Qwen 32B | 88.53% ± 0.35 | 83.09% ± 0.48 | 78.47% ± 0.59 | 88.60% ± 0.39 | 86.11% ± 0.64 | 88.83% ± 0.38 |
| Qwen 72B | **90.53%** ± **0.26** | 84.29% ± 0.44 | 80.52% ± 0.52 | 88.76% ± 0.38 | 84.71% ± 0.71 | 89.31% ± 0.37 |
| Llama 8B | 85.94% ± 0.44 | **88.82%** ± **0.26** | 83.02% ± 0.44 | 85.39% ± 0.50 | 80.45% ± 0.82 | 86.53% ± 0.46 |
| Llama 70B | 81.64% ± 0.59 | 82.34% ± 0.50 | **88.31%** ± **0.25** | 79.32% ± 0.69 | 70.17% ± 1.07 | 81.10% ± 0.65 |
| DeepSeekV3 | 88.17% ± 0.36 | 83.49% ± 0.47 | 77.81% ± 0.60 | **91.09%** ± **0.27** | 87.56% ± 0.61 | 89.26% ± 0.37 |
| GPT-4o Mini | 81.99% ± 0.64 | 77.21% ± 0.73 | 68.32% ± 0.90 | 85.14% ± 0.57 | **94.07%** ± **0.26** | 83.58% ± 0.65 |
| GPT-4o | 88.46% ± 0.35 | 84.30% ± 0.44 | 79.83% ± 0.55 | 88.87% ± 0.38 | 85.57% ± 0.70 | **91.67%** ± **0.26** |

Table 8: Accuracy of proxy LIME explanations on the text classification task: each value shows how well LIME explanations generated by the model on the **left** serve as surrogates for predicting the behavior of the model on the **top**.

|  | Qwen 0.5B | Qwen 1.5B | Qwen 3B | Qwen 7B | Qwen 14B | Qwen 32B |
|---|---|---|---|---|---|---|
| Qwen 0.5B | **98.55%** ± **0.16** | 41.38% ± 1.18 | 19.29% ± 0.87 | 24.61% ± 1.03 | 18.27% ± 0.87 | 26.38% ± 1.04 |
| Qwen 1.5B | 79.40% ± 0.90 | **60.14%** ± **0.81** | 39.43% ± 0.81 | 44.50% ± 0.85 | 38.46% ± 0.83 | 46.20% ± 0.84 |
| Qwen 3B | 28.79% ± 1.25 | 75.00% ± 0.59 | **81.69%** ± **0.77** | 82.12% ± 0.69 | 81.05% ± 0.82 | 80.75% ± 0.69 |
| Qwen 7B | 23.17% ± 0.98 | 73.51% ± 0.60 | 82.98% ± 0.63 | **84.77%** ± **0.54** | 83.98% ± 0.63 | 82.25% ± 0.57 |
| Qwen 14B | 23.21% ± 0.98 | 73.13% ± 0.60 | 82.83% ± 0.64 | 83.51% ± 0.58 | **84.77%** ± **0.61** | 82.58% ± 0.57 |
| Qwen 32B | 24.43% ± 0.98 | 73.50% ± 0.58 | 82.15% ± 0.65 | 83.00% ± 0.58 | 83.48% ± 0.64 | **83.15%** ± **0.55** |
| Qwen 72B | 25.17% ± 1.02 | 74.30% ± 0.57 | 82.26% ± 0.66 | 83.49% ± 0.58 | 83.34% ± 0.66 | 82.80% ± 0.56 |
| Llama 8B | 24.91% ± 0.98 | 74.01% ± 0.58 | 82.14% ± 0.63 | 83.21% ± 0.57 | 82.91% ± 0.64 | 81.62% ± 0.58 |
| Llama 70B | 30.06% ± 1.13 | 75.69% ± 0.54 | 79.68% ± 0.73 | 81.74% ± 0.63 | 80.22% ± 0.75 | 80.99% ± 0.62 |
| DeepSeekV3 | 71.42% ± 0.96 | 61.57% ± 0.66 | 46.02% ± 0.70 | 51.04% ± 0.69 | 45.51% ± 0.72 | 52.52% ± 0.68 |
| GPT-4o Mini | 33.67% ± 1.50 | 74.80% ± 0.65 | 77.26% ± 1.00 | 79.57% ± 0.86 | 77.73% ± 1.03 | 78.67% ± 0.85 |
| GPT-4o | 24.78% ± 0.99 | 73.74% ± 0.58 | 82.08% ± 0.65 | 83.38% ± 0.57 | 83.19% ± 0.65 | 82.15% ± 0.57 |

|  | Qwen 72B | Llama 8B | Llama 70B | DeepSeekV3 | GPT-4o Mini | GPT-4o |
|---|---|---|---|---|---|---|
| Qwen 0.5B | 28.16% ± 1.11 | 32.93% ± 1.13 | 42.30% ± 1.26 | 24.48% ± 1.03 | 12.76% ± 0.72 | 26.42% ± 1.10 |
| Qwen 1.5B | 47.97% ± 0.88 | 51.83% ± 0.85 | 60.81% ± 0.87 | 44.47% ± 0.85 | 33.12% ± 0.83 | 46.36% ± 0.86 |
| Qwen 3B | 81.06% ± 0.67 | 79.56% ± 0.60 | 75.97% ± 0.59 | 81.51% ± 0.72 | 79.82% ± 0.94 | 82.56% ± 0.67 |
| Qwen 7B | 82.14% ± 0.57 | 79.53% ± 0.56 | 74.39% ± 0.62 | 83.15% ± 0.59 | 83.57% ± 0.73 | 84.08% ± 0.56 |
| Qwen 14B | 82.17% ± 0.57 | 79.10% ± 0.57 | 74.03% ± 0.62 | 83.07% ± 0.59 | 83.63% ± 0.73 | 83.75% ± 0.57 |
| Qwen 32B | 82.30% ± 0.57 | 78.96% ± 0.57 | 74.62% ± 0.60 | 82.89% ± 0.59 | 82.48% ± 0.74 | 83.70% ± 0.57 |
| Qwen 72B | **83.44%** ± **0.54** | 79.96% ± 0.54 | 75.86% ± 0.56 | 83.53% ± 0.58 | 82.33% ± 0.76 | 84.29% ± 0.55 |
| Llama 8B | 81.87% ± 0.57 | **81.41%** ± **0.49** | 75.59% ± 0.57 | 82.65% ± 0.58 | 82.34% ± 0.73 | 83.43% ± 0.56 |
| Llama 70B | 81.66% ± 0.59 | 80.40% ± 0.53 | **78.79%** ± **0.49** | 81.72% ± 0.64 | 78.57% ± 0.86 | 83.04% ± 0.59 |
| DeepSeekV3 | 54.46% ± 0.71 | 56.81% ± 0.66 | 64.57% ± 0.67 | **51.67%** ± **0.71** | 40.64% ± 0.76 | 53.12% ± 0.69 |
| GPT-4o Mini | 79.51% ± 0.82 | 78.58% ± 0.71 | 76.89% ± 0.64 | 79.11% ± 0.89 | **76.38%** ± **1.18** | 80.69% ± 0.82 |
| GPT-4o | 82.18% ± 0.56 | 79.33% ± 0.56 | 74.98% ± 0.59 | 83.01% ± 0.58 | 82.45% ± 0.75 | **84.63%** ± **0.54** |

Table 9: Accuracy of proxy Kernel SHAP explanations on the text classification task: each value shows how well Kernel SHAP explanations generated by the model on the **left** serve as surrogates for predicting the behavior of the model on the **top**.

|  | Qwen 0.5B | Qwen 1.5B | Qwen 3B | Qwen 7B | Qwen 14B | Qwen 32B |
|---|---|---|---|---|---|---|
| Qwen 0.5B | **88.40% ± 1.21** | 57.69% ± 3.96 | 54.85% ± 4.04 | 54.68% ± 4.00 | 52.45% ± 3.82 | 52.60% ± 4.07 |
| Qwen 1.5B | 57.69% ± 3.72 | **87.85% ± 1.19** | 66.16% ± 3.41 | 62.99% ± 3.57 | 62.17% ± 3.46 | 60.66% ± 3.84 |
| Qwen 3B | 53.33% ± 4.11 | 66.66% ± 3.28 | **87.50% ± 1.18** | 66.64% ± 3.30 | 67.46% ± 3.23 | 66.13% ± 3.72 |
| Qwen 7B | 51.98% ± 4.20 | 62.08% ± 3.75 | 66.84% ± 3.50 | **87.32% ± 1.20** | 71.58% ± 3.05 | 68.92% ± 3.59 |
| Qwen 14B | 51.61% ± 4.03 | 60.55% ± 3.65 | 66.29% ± 3.39 | 68.77% ± 3.27 | **86.26% ± 1.28** | 73.70% ± 3.08 |
| Qwen 32B | 52.17% ± 4.19 | 58.59% ± 3.99 | 65.03% ± 3.61 | 66.96% ± 3.42 | 72.10% ± 3.02 | **88.61% ± 1.20** |
| Qwen 72B | 53.53% ± 4.19 | 59.96% ± 3.79 | 63.63% ± 3.65 | 67.55% ± 3.39 | 69.91% ± 3.09 | 75.94% ± 3.11 |
| Llama 8B | 50.80% ± 3.93 | 61.74% ± 3.72 | 64.80% ± 3.29 | 66.87% ± 3.31 | 65.83% ± 3.20 | 64.98% ± 3.59 |
| Llama 70B | 53.71% ± 4.05 | 60.43% ± 3.89 | 64.73% ± 3.62 | 68.63% ± 3.25 | 69.92% ± 3.13 | 74.35% ± 3.07 |
| DeepSeekV3 | 51.13% ± 4.21 | 58.08% ± 3.92 | 63.28% ± 3.56 | 67.76% ± 3.37 | 70.82% ± 3.05 | 76.40% ± 2.83 |
| GPT-4o Mini | 51.46% ± 4.05 | 62.03% ± 3.75 | 65.46% ± 3.45 | 65.75% ± 3.47 | 70.48% ± 3.03 | 73.60% ± 3.10 |
| GPT-4o | 52.82% ± 4.21 | 57.45% ± 3.77 | 62.72% ± 3.68 | 64.12% ± 3.53 | 68.97% ± 3.08 | 72.44% ± 3.33 |

|  | Qwen 72B | Llama 8B | Llama 70B | DeepSeekV3 | GPT-4o Mini | GPT-4o |
|---|---|---|---|---|---|---|
| Qwen 0.5B | 52.58% ± 4.19 | 51.61% ± 3.77 | 53.97% ± 3.96 | 50.99% ± 4.16 | 52.96% ± 3.97 | 49.53% ± 4.38 |
| Qwen 1.5B | 59.95% ± 3.88 | 60.28% ± 3.54 | 61.35% ± 3.69 | 58.04% ± 3.91 | 60.91% ± 3.74 | 54.69% ± 4.35 |
| Qwen 3B | 63.84% ± 3.84 | 64.41% ± 3.27 | 66.42% ± 3.48 | 64.77% ± 3.63 | 64.92% ± 3.57 | 57.94% ± 4.43 |
| Qwen 7B | 70.98% ± 3.49 | 66.58% ± 3.29 | 70.46% ± 3.36 | 70.82% ± 3.30 | 65.65% ± 3.76 | 58.59% ± 4.68 |
| Qwen 14B | 73.27% ± 3.13 | 63.13% ± 3.46 | 70.42% ± 3.20 | 73.70% ± 2.96 | 69.61% ± 3.24 | 64.58% ± 3.94 |
| Qwen 32B | 76.10% ± 3.20 | 61.95% ± 3.55 | 73.08% ± 3.20 | 77.07% ± 2.80 | 70.42% ± 3.46 | 67.63% ± 4.15 |
| Qwen 72B | **90.10% ± 1.10** | 60.33% ± 3.66 | 73.63% ± 3.09 | 75.14% ± 3.08 | 73.16% ± 3.14 | 71.58% ± 3.78 |
| Llama 8B | 63.84% ± 3.67 | **85.70% ± 1.20** | 67.60% ± 3.33 | 65.18% ± 3.48 | 64.61% ± 3.58 | 55.19% ± 4.38 |
| Llama 70B | 74.00% ± 3.09 | 64.75% ± 3.37 | **87.69% ± 1.18** | 72.60% ± 3.19 | 70.56% ± 3.39 | 65.14% ± 4.37 |
| DeepSeekV3 | 74.51% ± 3.27 | 61.24% ± 3.64 | 71.39% ± 3.24 | **88.42% ± 1.25** | 70.83% ± 3.33 | 68.90% ± 3.99 |
| GPT-4o Mini | 75.06% ± 3.10 | 63.59% ± 3.49 | 73.85% ± 2.97 | 73.68% ± 3.09 | **85.47% ± 1.73** | 71.61% ± 3.48 |
| GPT-4o | 74.68% ± 3.23 | 59.56% ± 3.63 | 70.98% ± 3.30 | 75.32% ± 3.00 | 71.39% ± 3.16 | **80.39% ± 2.61** |

Table 10: Accuracy of proxy LIME explanations on high school chemistriy of MMLU datasets: each value shows how well LIME explanations generated by the model on the **left** serve as surrogates for predicting the behavior of the model on the **top**.

## G.2 MULTIPLE-CHOICE QUESTION ANSWERING

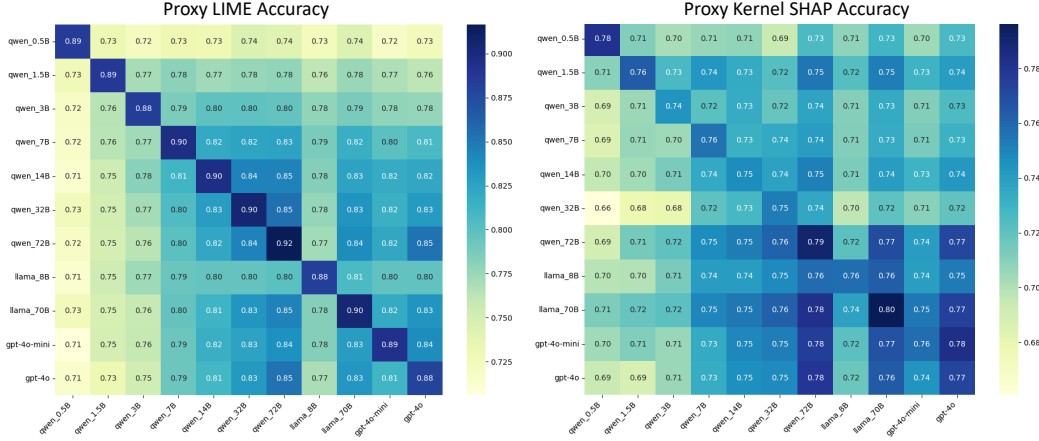

Figure 4: Accuracy of LIME proxy explanations on the multiple-choice question answering task. Each cell shows how well explanations generated by the model on the **y-axis** serve as surrogates for predicting the behavior of the model on the **x-axis**. The heatmap on the right shows results after filtering out examples where the budget-friendly and expensive models produce different predictions for the input.

We have provided overall fidelity results in Figure 4 and 5, and we provide also the explanation fidelity results of Kernel SHAP in Figure 5, and on each subject in Figure 6 and Figure 7.

We can see the observation we find in section 4 also holds for Kernel SHAP and each subject, i.e., filtering out examples where the budget-friendly and expensive models produce different predictions for the same input can significantly improve the fidelity of proxy explanations. Additionally, we also

|  | Qwen 0.5B | Qwen 1.5B | Qwen 3B | Qwen 7B | Qwen 14B | Qwen 32B |
|---|---|---|---|---|---|---|
| Qwen 0.5B | **88.40%** ± **1.21** | 66.96% ± 5.07 | 68.54% ± 5.25 | 66.02% ± 5.54 | 66.96% ± 6.10 | 70.93% ± 5.80 |
| Qwen 1.5B | 68.20% ± 4.93 | **87.85%** ± **1.19** | 72.08% ± 3.90 | 70.61% ± 4.30 | 69.95% ± 4.25 | 70.82% ± 4.89 |
| Qwen 3B | 69.24% ± 5.44 | 71.96% ± 4.18 | **87.50%** ± **1.18** | 72.88% ± 3.84 | 74.31% ± 3.90 | 75.15% ± 4.18 |
| Qwen 7B | 65.19% ± 5.96 | 69.80% ± 4.61 | 72.27% ± 3.87 | **87.32%** ± **1.20** | 76.26% ± 3.43 | 76.91% ± 3.71 |
| Qwen 14B | 66.06% ± 6.58 | 68.60% ± 4.57 | 73.32% ± 3.84 | 75.29% ± 3.47 | **86.26%** ± **1.28** | 77.33% ± 3.27 |
| Qwen 32B | 70.37% ± 6.29 | 68.36% ± 5.19 | 72.64% ± 4.19 | 74.01% ± 3.68 | 75.40% ± 3.27 | **88.61%** ± **1.20** |
| Qwen 72B | 71.14% ± 5.77 | 69.95% ± 4.55 | 71.34% ± 4.17 | 74.25% ± 3.53 | 75.85% ± 3.22 | 79.72% ± 3.06 |
| Llama 8B | 66.86% ± 6.08 | 70.93% ± 4.59 | 71.45% ± 3.90 | 73.62% ± 3.67 | 73.00% ± 3.51 | 73.83% ± 3.68 |
| Llama 70B | 73.26% ± 5.44 | 70.58% ± 4.71 | 70.97% ± 4.28 | 74.65% ± 3.81 | 74.30% ± 3.56 | 78.56% ± 2.97 |
| DeepSeekV3 | 68.55% ± 6.21 | 66.40% ± 5.09 | 71.05% ± 4.44 | 75.06% ± 3.47 | 76.81% ± 3.26 | 79.35% ± 2.76 |
| GPT-4o Mini | 68.20% ± 5.87 | 70.35% ± 4.74 | 71.08% ± 3.81 | 73.10% ± 4.00 | 76.30% ± 3.46 | 79.71% ± 3.14 |
| GPT-4o | 71.88% ± 5.40 | 64.87% ± 4.72 | 71.27% ± 4.24 | 73.99% ± 3.63 | 74.42% ± 3.52 | 79.07% ± 3.19 |

|  | Qwen 72B | Llama 8B | Llama 70B | DeepSeekV3 | GPT-4o Mini | GPT-4o |
|---|---|---|---|---|---|---|
| Qwen 0.5B | 70.60% ± 5.61 | 67.99% ± 5.86 | 73.40% ± 5.29 | 69.92% ± 5.95 | 66.09% ± 5.76 | 69.36% ± 5.97 |
| Qwen 1.5B | 72.49% ± 4.45 | 70.68% ± 4.27 | 72.49% ± 4.44 | 69.08% ± 4.80 | 71.08% ± 4.65 | 67.79% ± 4.84 |
| Qwen 3B | 73.28% ± 4.29 | 71.73% ± 3.75 | 73.36% ± 4.24 | 73.85% ± 4.31 | 70.25% ± 4.09 | 70.51% ± 4.75 |
| Qwen 7B | 77.16% ± 3.65 | 72.95% ± 3.77 | 76.81% ± 3.75 | 77.72% ± 3.55 | 72.81% ± 4.21 | 73.42% ± 4.40 |
| Qwen 14B | 78.43% ± 3.17 | 71.18% ± 3.60 | 75.27% ± 3.53 | 79.44% ± 3.17 | 75.43% ± 3.74 | 73.76% ± 4.00 |
| Qwen 32B | 80.01% ± 3.12 | 70.98% ± 3.73 | 77.16% ± 3.01 | 79.91% ± 2.80 | 76.83% ± 3.39 | 77.47% ± 3.48 |
| Qwen 72B | **90.10%** ± **1.10** | 70.20% ± 3.66 | 77.73% ± 3.11 | 79.11% ± 3.07 | 76.91% ± 3.34 | 80.03% ± 3.01 |
| Llama 8B | 73.49% ± 3.76 | **85.70%** ± **1.20** | 74.47% ± 3.63 | 73.75% ± 3.66 | 73.98% ± 3.72 | 72.32% ± 3.99 |
| Llama 70B | 78.24% ± 3.04 | 71.99% ± 3.72 | **87.69%** ± **1.18** | 76.71% ± 3.36 | 77.53% ± 3.22 | 76.71% ± 3.66 |
| DeepSeekV3 | 78.48% ± 3.25 | 70.90% ± 3.66 | 75.12% ± 3.35 | **88.42%** ± **1.25** | 76.31% ± 3.43 | 77.03% ± 3.40 |
| GPT-4o Mini | 79.15% ± 3.22 | 72.64% ± 3.70 | 79.25% ± 2.92 | 78.83% ± 3.34 | **85.47%** ± **1.73** | 77.58% ± 3.36 |
| GPT-4o | 80.69% ± 2.98 | 70.96% ± 3.82 | 78.01% ± 3.05 | 80.09% ± 2.90 | 73.96% ± 3.47 | **80.39%** ± **2.61** |

Table 11: Accuracy of proxy **filtered** LIME explanations on high school chemistry of MMLU datasets: each value shows how well LIME explanations generated by the model on the **left** serve as surrogates for predicting the behavior of the model on the **top**.

|  | Qwen 0.5B | Qwen 1.5B | Qwen 3B | Qwen 7B | Qwen 14B | Qwen 32B |
|---|---|---|---|---|---|---|
| Qwen 0.5B | **87.75%** ± **1.77** | 64.64% ± 5.28 | 64.23% ± 4.75 | 63.54% ± 5.23 | 61.44% ± 5.29 | 61.47% ± 5.26 |
| Qwen 1.5B | 65.08% ± 5.40 | **87.13%** ± **1.75** | 73.00% ± 4.29 | 73.94% ± 4.41 | 71.53% ± 4.56 | 70.29% ± 4.87 |
| Qwen 3B | 62.31% ± 5.32 | 71.57% ± 4.48 | **86.11%** ± **1.81** | 76.87% ± 3.56 | 76.13% ± 3.51 | 75.05% ± 4.09 |
| Qwen 7B | 61.25% ± 5.58 | 71.35% ± 4.54 | 72.84% ± 4.20 | **88.08%** ± **1.71** | 79.37% ± 3.53 | 81.29% ± 3.20 |
| Qwen 14B | 59.28% ± 5.70 | 68.95% ± 4.72 | 74.12% ± 3.50 | 78.41% ± 3.68 | **88.21%** ± **1.73** | 81.78% ± 3.56 |
| Qwen 32B | 62.31% ± 5.70 | 69.15% ± 5.00 | 73.90% ± 3.69 | 78.19% ± 3.69 | 80.13% ± 3.65 | **88.88%** ± **1.76** |
| Qwen 72B | 60.20% ± 5.70 | 68.31% ± 4.99 | 73.58% ± 3.78 | 78.79% ± 3.39 | 78.75% ± 3.71 | 83.34% ± 2.98 |
| Llama 8B | 59.99% ± 5.54 | 68.22% ± 4.68 | 72.18% ± 3.75 | 73.98% ± 4.23 | 73.72% ± 4.14 | 71.93% ± 4.70 |
| Llama 70B | 58.14% ± 5.65 | 66.28% ± 5.00 | 71.51% ± 4.37 | 77.35% ± 3.70 | 77.30% ± 4.02 | 80.73% ± 3.51 |
| DeepSeekV3 | 59.53% ± 5.73 | 66.98% ± 5.06 | 69.21% ± 4.49 | 78.17% ± 3.44 | 77.44% ± 3.88 | 79.76% ± 3.66 |
| GPT-4o Mini | 59.05% ± 5.77 | 68.49% ± 4.76 | 72.46% ± 4.12 | 77.73% ± 3.65 | 76.53% ± 4.06 | 77.04% ± 4.18 |
| GPT-4o | 56.87% ± 5.83 | 65.23% ± 5.27 | 68.73% ± 4.49 | 75.43% ± 3.69 | 77.01% ± 3.86 | 78.82% ± 3.61 |

|  | Qwen 72B | Llama 8B | Llama 70B | DeepSeekV3 | GPT-4o Mini | GPT-4o |
|---|---|---|---|---|---|---|
| Qwen 0.5B | 60.62% ± 5.42 | 64.07% ± 4.89 | 60.28% ± 5.42 | 62.83% ± 5.49 | 60.11% ± 5.55 | 57.56% ± 5.88 |
| Qwen 1.5B | 69.01% ± 4.93 | 69.60% ± 4.66 | 68.53% ± 5.05 | 70.21% ± 4.99 | 69.42% ± 4.95 | 67.86% ± 5.52 |
| Qwen 3B | 74.92% ± 3.94 | 73.27% ± 3.85 | 72.48% ± 4.47 | 74.65% ± 4.14 | 73.18% ± 4.34 | 71.19% ± 4.95 |
| Qwen 7B | 79.54% ± 3.71 | 72.38% ± 4.34 | 78.66% ± 3.73 | 80.10% ± 3.56 | 76.07% ± 4.16 | 76.38% ± 4.24 |
| Qwen 14B | 79.38% ± 3.85 | 72.38% ± 4.14 | 77.14% ± 4.07 | 79.62% ± 3.84 | 75.57% ± 4.47 | 76.00% ± 4.78 |
| Qwen 32B | 82.74% ± 3.23 | 72.57% ± 4.52 | 78.99% ± 3.98 | 80.33% ± 3.87 | 74.80% ± 4.71 | 74.89% ± 4.75 |
| Qwen 72B | **88.95%** ± **1.67** | 71.99% ± 4.36 | 79.98% ± 3.45 | 79.87% ± 4.03 | 77.39% ± 4.09 | 78.00% ± 4.29 |
| Llama 8B | 73.12% ± 4.52 | **85.79%** ± **1.75** | 72.08% ± 4.73 | 73.19% ± 4.55 | 73.25% ± 4.47 | 71.36% ± 5.12 |
| Llama 70B | 80.27% ± 3.58 | 70.69% ± 4.71 | **87.45%** ± **2.10** | 77.40% ± 4.43 | 74.81% ± 4.53 | 75.62% ± 4.76 |
| DeepSeekV3 | 77.38% ± 4.29 | 69.87% ± 4.60 | 76.71% ± 4.28 | **87.32%** ± **2.21** | 76.09% ± 4.56 | 76.09% ± 4.56 |
| GPT-4o Mini | 78.12% ± 4.03 | 71.51% ± 4.57 | 75.75% ± 4.20 | 77.36% ± 4.45 | **87.03%** ± **2.25** | 80.57% ± 4.04 |
| GPT-4o | 78.76% ± 3.66 | 68.20% ± 4.56 | 78.22% ± 3.46 | 78.18% ± 3.86 | 76.11% ± 4.20 | **82.58%** ± **3.51** |

Table 12: Accuracy of proxy LIME explanations on high school computer science of MMLU datasets: each value shows how well LIME explanations generated by the model on the **left** serve as surrogates for predicting the behavior of the model on the **top**.

| | Qwen 0.5B | Qwen 1.5B | Qwen 3B | Qwen 7B | Qwen 14B | Qwen 32B |
|---|---|---|---|---|---|---|
| Qwen 0.5B | **87.75%** ± **1.77** | 75.97% ± 5.91 | 74.61% ± 5.48 | 74.28% ± 6.82 | 75.35% ± 6.90 | 78.50% ± 6.19 |
| Qwen 1.5B | 75.29% ± 5.73 | **87.13%** ± **1.75** | 77.29% ± 4.67 | 80.88% ± 4.18 | 79.97% ± 4.45 | 81.58% ± 4.05 |
| Qwen 3B | 73.31% ± 5.61 | 76.74% ± 4.37 | **86.11%** ± **1.81** | 80.15% ± 3.70 | 80.01% ± 3.81 | 81.41% ± 3.90 |
| Qwen 7B | 72.53% ± 6.77 | 80.23% ± 3.89 | 79.53% ± 3.88 | **88.08%** ± **1.71** | 82.60% ± 3.49 | 82.34% ± 3.39 |
| Qwen 14B | 72.79% ± 7.00 | 78.21% ± 4.43 | 78.82% ± 3.96 | 81.21% ± 3.69 | **88.21%** ± **1.73** | 84.08% ± 3.26 |
| Qwen 32B | 75.92% ± 6.52 | 79.77% ± 4.14 | 78.79% ± 4.17 | 79.52% ± 3.72 | 82.13% ± 3.48 | **88.88%** ± **1.76** |
| Qwen 72B | 72.53% ± 7.12 | 77.10% ± 4.61 | 79.00% ± 3.99 | 80.20% ± 3.59 | 80.74% ± 3.73 | 83.79% ± 3.00 |
| Llama 8B | 75.51% ± 6.51 | 75.91% ± 4.53 | 77.24% ± 3.88 | 78.13% ± 4.38 | 78.28% ± 4.35 | 78.50% ± 4.63 |
| Llama 70B | 71.92% ± 6.93 | 75.73% ± 4.71 | 76.56% ± 4.73 | 78.46% ± 4.00 | 78.79% ± 4.19 | 81.30% ± 3.65 |
| DeepSeekV3 | 72.91% ± 6.81 | 77.23% ± 4.76 | 76.21% ± 4.67 | 79.15% ± 3.70 | 80.15% ± 3.63 | 81.15% ± 3.50 |
| GPT-4o Mini | 69.10% ± 7.67 | 76.07% ± 4.74 | 74.98% ± 4.58 | 78.29% ± 4.04 | 77.98% ± 4.68 | 79.63% ± 4.57 |
| GPT-4o | 68.86% ± 7.57 | 76.25% ± 4.76 | 76.19% ± 4.36 | 78.01% ± 3.76 | 80.03% ± 4.01 | 81.12% ± 3.56 |

| | Qwen 72B | Llama 8B | Llama 70B | DeepSeekV3 | GPT-4o Mini | GPT-4o |
|---|---|---|---|---|---|---|
| Qwen 0.5B | 76.28% ± 6.99 | 75.93% ± 5.83 | 75.21% ± 7.04 | 75.81% ± 6.80 | 71.57% ± 7.71 | 70.18% ± 8.21 |
| Qwen 1.5B | 79.39% ± 4.55 | 75.18% ± 4.75 | 78.71% ± 4.74 | 79.69% ± 4.79 | 78.06% ± 4.97 | 78.48% ± 4.95 |
| Qwen 3B | 81.80% ± 3.83 | 75.92% ± 3.95 | 78.62% ± 4.57 | 79.74% ± 4.43 | 77.98% ± 4.41 | 77.99% ± 4.81 |
| Qwen 7B | 82.25% ± 3.60 | 76.83% ± 4.44 | 81.41% ± 3.74 | 81.49% ± 3.80 | 79.14% ± 4.20 | 78.80% ± 4.43 |
| Qwen 14B | 82.50% ± 3.53 | 76.25% ± 4.29 | 79.61% ± 4.06 | 81.77% ± 3.77 | 78.62% ± 4.79 | 79.35% ± 4.72 |
| Qwen 32B | 83.49% ± 3.25 | 76.69% ± 4.56 | 81.05% ± 3.72 | 81.65% ± 3.77 | 78.80% ± 4.93 | 77.88% ± 4.60 |
| Qwen 72B | **88.95%** ± **1.67** | 76.58% ± 4.39 | 81.47% ± 3.34 | 80.65% ± 3.98 | 80.23% ± 4.12 | 80.59% ± 3.97 |
| Llama 8B | 79.71% ± 4.31 | **85.79%** ± **1.75** | 77.65% ± 4.83 | 79.62% ± 4.37 | 79.11% ± 4.31 | 78.94% ± 4.84 |
| Llama 70B | 81.61% ± 3.44 | 74.97% ± 4.77 | **87.45%** ± **2.10** | 78.72% ± 4.51 | 78.01% ± 4.70 | 78.94% ± 4.43 |
| DeepSeekV3 | 79.27% ± 4.08 | 75.64% ± 4.38 | 78.01% ± 4.43 | **87.32%** ± **2.21** | 77.47% ± 4.98 | 77.82% ± 4.52 |
| GPT-4o Mini | 79.87% ± 4.31 | 76.12% ± 4.51 | 77.77% ± 4.42 | 79.18% ± 4.67 | **87.03%** ± **2.25** | 81.47% ± 4.24 |
| GPT-4o | 81.72% ± 3.38 | 74.17% ± 4.64 | 80.52% ± 3.56 | 79.88% ± 3.92 | 78.63% ± 4.44 | **82.58%** ± **3.51** |

Table 13: Accuracy of proxy **filtered** LIME explanations on high school computer science of MMLU datasets: each value shows how well LIME explanations generated by the model on the **left** serve as surrogates for predicting the behavior of the model on the **top**.

| | Qwen 0.5B | Qwen 1.5B | Qwen 3B | Qwen 7B | Qwen 14B | Qwen 32B |
|---|---|---|---|---|---|---|
| Qwen 0.5B | **87.93%** ± **1.08** | 61.79% ± 3.38 | 58.21% ± 3.36 | 55.88% ± 3.79 | 55.37% ± 3.68 | 53.39% ± 3.77 |
| Qwen 1.5B | 59.72% ± 3.53 | **88.26%** ± **1.06** | 70.39% ± 2.95 | 68.05% ± 3.40 | 66.76% ± 3.38 | 65.19% ± 3.53 |
| Qwen 3B | 57.43% ± 3.60 | 69.15% ± 3.10 | **87.87%** ± **1.17** | 73.62% ± 3.04 | 74.55% ± 2.83 | 71.28% ± 3.03 |
| Qwen 7B | 56.09% ± 3.74 | 65.12% ± 3.44 | 71.40% ± 3.02 | **89.77%** ± **1.11** | 79.03% ± 2.51 | 80.10% ± 2.36 |
| Qwen 14B | 55.36% ± 3.79 | 64.97% ± 3.48 | 72.78% ± 2.86 | 79.30% ± 2.52 | **89.15%** ± **1.13** | 82.18% ± 2.19 |
| Qwen 32B | 54.31% ± 3.81 | 62.69% ± 3.57 | 69.43% ± 3.03 | 79.95% ± 2.34 | 80.79% ± 2.28 | **89.61%** ± **1.18** |
| Qwen 72B | 54.47% ± 3.78 | 62.55% ± 3.58 | 67.56% ± 3.12 | 77.28% ± 2.72 | 79.18% ± 2.51 | 82.41% ± 2.12 |
| Llama 8B | 57.24% ± 3.51 | 67.74% ± 3.05 | 70.65% ± 2.93 | 72.40% ± 3.22 | 71.27% ± 3.16 | 70.09% ± 3.26 |
| Llama 70B | 54.14% ± 3.76 | 64.38% ± 3.55 | 68.67% ± 3.10 | 77.43% ± 2.65 | 78.06% ± 2.63 | 80.20% ± 2.49 |
| DeepSeekV3 | 53.42% ± 3.90 | 61.95% ± 3.67 | 67.78% ± 3.29 | 77.37% ± 2.77 | 78.74% ± 2.63 | 82.08% ± 2.21 |
| GPT-4o Mini | 53.96% ± 3.71 | 65.55% ± 3.33 | 70.40% ± 2.96 | 76.70% ± 2.70 | 78.93% ± 2.47 | 79.09% ± 2.48 |
| GPT-4o | 54.86% ± 3.86 | 62.63% ± 3.55 | 67.03% ± 3.31 | 78.17% ± 2.61 | 77.30% ± 2.74 | 80.74% ± 2.44 |

| | Qwen 72B | Llama 8B | Llama 70B | DeepSeekV3 | GPT-4o Mini | GPT-4o |
|---|---|---|---|---|---|---|
| Qwen 0.5B | 52.72% ± 3.98 | 57.12% ± 3.60 | 54.63% ± 3.81 | 52.68% ± 4.03 | 55.24% ± 3.67 | 52.17% ± 3.98 |
| Qwen 1.5B | 64.62% ± 3.63 | 69.07% ± 3.03 | 67.05% ± 3.46 | 63.84% ± 3.72 | 67.33% ± 3.25 | 64.29% ± 3.56 |
| Qwen 3B | 70.11% ± 3.30 | 70.33% ± 3.11 | 70.82% ± 3.24 | 68.82% ± 3.47 | 71.80% ± 3.07 | 69.00% ± 3.39 |
| Qwen 7B | 79.31% ± 2.69 | 70.79% ± 3.17 | 77.66% ± 2.72 | 79.16% ± 2.77 | 76.56% ± 2.75 | 78.89% ± 2.76 |
| Qwen 14B | 81.44% ± 2.57 | 70.45% ± 3.15 | 79.91% ± 2.58 | 81.54% ± 2.52 | 79.55% ± 2.45 | 79.87% ± 2.67 |
| Qwen 32B | 83.54% ± 2.29 | 69.52% ± 3.13 | 81.12% ± 2.51 | 83.73% ± 2.23 | 78.92% ± 2.48 | 82.13% ± 2.38 |
| Qwen 72B | **91.30%** ± **1.03** | 67.88% ± 3.22 | 80.94% ± 2.56 | 85.08% ± 2.11 | 78.87% ± 2.53 | 84.04% ± 2.21 |
| Llama 8B | 68.56% ± 3.39 | **88.01%** ± **1.17** | 73.17% ± 3.05 | 69.80% ± 3.51 | 69.99% ± 3.23 | 69.48% ± 3.39 |
| Llama 70B | 81.23% ± 2.64 | 71.47% ± 3.04 | **90.14%** ± **1.18** | 80.90% ± 2.68 | 78.95% ± 2.61 | 81.89% ± 2.47 |
| DeepSeekV3 | 84.24% ± 2.24 | 68.40% ± 3.33 | 79.26% ± 2.77 | **91.10%** ± **1.26** | 78.07% ± 2.76 | 83.29% ± 2.38 |
| GPT-4o Mini | 80.84% ± 2.57 | 69.68% ± 3.16 | 80.13% ± 2.58 | 79.90% ± 2.69 | **88.90%** ± **1.20** | 82.35% ± 2.31 |
| GPT-4o | 83.78% ± 2.33 | 69.32% ± 3.23 | 80.61% ± 2.55 | 84.00% ± 2.27 | 79.43% ± 2.49 | **89.74%** ± **1.35** |

Table 14: Accuracy of proxy LIME explanations on high school microeconomics of MMLU datasets: each value shows how well LIME explanations generated by the model on the **left** serve as surrogates for predicting the behavior of the model on the **top**.

| | Qwen 0.5B | Qwen 1.5B | Qwen 3B | Qwen 7B | Qwen 14B | Qwen 32B |
|---|---|---|---|---|---|---|
| Qwen 0.5B | **87.93% ± 1.08** | 71.42% ± 3.82 | 70.66% ± 3.94 | 75.31% ± 3.72 | 71.32% ± 4.38 | 72.11% ± 4.26 |
| Qwen 1.5B | 72.28% ± 3.80 | **88.26% ± 1.06** | 76.60% ± 2.96 | 77.90% ± 2.79 | 75.49% ± 3.09 | 74.98% ± 3.17 |
| Qwen 3B | 71.68% ± 4.06 | 75.48% ± 3.01 | **87.87% ± 1.17** | 78.61% ± 2.86 | 79.81% ± 2.63 | 77.66% ± 2.90 |
| Qwen 7B | 74.44% ± 3.93 | 74.49% ± 3.07 | 75.93% ± 3.03 | **89.77% ± 1.11** | 81.19% ± 2.52 | 82.64% ± 2.21 |
| Qwen 14B | 71.16% ± 4.62 | 73.29% ± 3.36 | 77.97% ± 2.79 | 81.01% ± 2.56 | **89.15% ± 1.13** | 83.69% ± 2.22 |
| Qwen 32B | 71.51% ± 4.42 | 71.62% ± 3.45 | 74.99% ± 3.08 | 82.05% ± 2.26 | 82.01% ± 2.35 | **89.61% ± 1.18** |
| Qwen 72B | 71.15% ± 4.35 | 73.10% ± 3.44 | 73.10% ± 3.10 | 80.02% ± 2.51 | 81.25% ± 2.32 | 83.41% ± 2.07 |
| Llama 8B | 69.67% ± 4.22 | 73.13% ± 2.98 | 77.13% ± 2.97 | 79.04% ± 2.70 | 78.07% ± 2.77 | 77.14% ± 2.89 |
| Llama 70B | 72.36% ± 4.08 | 72.67% ± 3.45 | 73.77% ± 3.20 | 79.87% ± 2.48 | 80.70% ± 2.45 | 82.09% ± 2.30 |
| DeepSeekV3 | 72.62% ± 4.39 | 72.14% ± 3.47 | 74.21% ± 3.28 | 80.56% ± 2.55 | 80.91% ± 2.47 | 83.19% ± 2.18 |
| GPT-4o Mini | 69.86% ± 4.43 | 73.41% ± 3.36 | 74.36% ± 3.09 | 79.30% ± 2.54 | 81.40% ± 2.31 | 81.06% ± 2.39 |
| GPT-4o | 72.41% ± 4.31 | 70.87% ± 3.55 | 72.07% ± 3.43 | 80.59% ± 2.52 | 79.33% ± 2.65 | 82.50% ± 2.26 |

| | Qwen 72B | Llama 8B | Llama 70B | DeepSeekV3 | GPT-4o Mini | GPT-4o |
|---|---|---|---|---|---|---|
| Qwen 0.5B | 73.13% ± 4.20 | 69.84% ± 4.17 | 72.93% ± 4.15 | 74.78% ± 4.25 | 70.34% ± 4.34 | 73.23% ± 4.25 |
| Qwen 1.5B | 75.76% ± 3.21 | 74.88% ± 2.90 | 75.81% ± 3.27 | 75.16% ± 3.25 | 75.07% ± 3.18 | 74.41% ± 3.21 |
| Qwen 3B | 77.03% ± 3.02 | 77.87% ± 2.94 | 76.86% ± 3.11 | 76.90% ± 3.21 | 76.24% ± 3.02 | 75.66% ± 3.23 |
| Qwen 7B | 82.62% ± 2.34 | 77.60% ± 2.78 | 80.77% ± 2.51 | 83.14% ± 2.33 | 79.40% ± 2.57 | 82.99% ± 2.30 |
| Qwen 14B | 84.06% ± 2.23 | 76.78% ± 2.95 | 82.49% ± 2.42 | 83.49% ± 2.39 | 81.62% ± 2.36 | 82.31% ± 2.44 |
| Qwen 32B | 85.46% ± 1.98 | 75.55% ± 3.00 | 82.86% ± 2.38 | 84.95% ± 2.10 | 80.35% ± 2.46 | 83.93% ± 2.13 |
| Qwen 72B | **91.30% ± 1.03** | 73.53% ± 3.15 | 83.95% ± 2.15 | 86.10% ± 2.01 | 81.21% ± 2.25 | 85.98% ± 1.78 |
| Llama 8B | 76.11% ± 3.05 | **88.01% ± 1.17** | 79.83% ± 2.70 | 78.65% ± 2.99 | 76.35% ± 3.08 | 77.54% ± 2.95 |
| Llama 70B | 84.99% ± 2.05 | 77.32% ± 2.79 | **90.14% ± 1.18** | 83.70% ± 2.30 | 81.08% ± 2.46 | 84.05% ± 2.21 |
| DeepSeekV3 | 85.85% ± 1.99 | 75.30% ± 3.15 | 82.38% ± 2.42 | **91.10% ± 1.26** | 79.90% ± 2.60 | 84.97% ± 2.03 |
| GPT-4o Mini | 83.76% ± 2.17 | 75.54% ± 3.05 | 82.48% ± 2.38 | 82.57% ± 2.39 | **88.90% ± 1.20** | 84.02% ± 2.17 |
| GPT-4o | 85.85% ± 1.96 | 75.12% ± 3.16 | 82.38% ± 2.44 | 85.23% ± 2.06 | 80.76% ± 2.45 | **89.74% ± 1.35** |

Table 15: Accuracy of proxy **filtered** LIME explanations on high school microeconomics of MMLU datasets: each value shows how well LIME explanations generated by the model on the **left** serve as surrogates for predicting the behavior of the model on the **top**.

| | Qwen 0.5B | Qwen 1.5B | Qwen 3B | Qwen 7B | Qwen 14B | Qwen 32B |
|---|---|---|---|---|---|---|
| Qwen 0.5B | **88.25% ± 0.64** | 66.76% ± 2.11 | 64.46% ± 2.06 | 63.40% ± 2.23 | 62.75% ± 2.29 | 61.03% ± 2.33 |
| Qwen 1.5B | 65.69% ± 2.10 | **88.57% ± 0.68** | 75.99% ± 1.57 | 77.51% ± 1.59 | 76.13% ± 1.75 | 75.68% ± 1.80 |
| Qwen 3B | 63.16% ± 2.12 | 75.93% ± 1.66 | **87.24% ± 0.71** | 78.53% ± 1.60 | 79.39% ± 1.56 | 78.77% ± 1.61 |
| Qwen 7B | 62.57% ± 2.21 | 75.22% ± 1.72 | 76.68% ± 1.60 | **89.29% ± 0.67** | 82.26% ± 1.40 | 82.27% ± 1.50 |
| Qwen 14B | 61.51% ± 2.25 | 73.82% ± 1.83 | 76.13% ± 1.63 | 81.10% ± 1.44 | **89.63% ± 0.68** | 85.26% ± 1.18 |
| Qwen 32B | 60.70% ± 2.26 | 72.87% ± 1.82 | 75.14% ± 1.61 | 80.36% ± 1.48 | 84.08% ± 1.21 | **89.88% ± 0.68** |
| Qwen 72B | 60.12% ± 2.34 | 72.42% ± 1.95 | 73.94% ± 1.72 | 79.51% ± 1.57 | 82.40% ± 1.39 | 84.50% ± 1.25 |
| Llama 8B | 61.59% ± 2.21 | 73.76% ± 1.83 | 76.16% ± 1.60 | 79.36% ± 1.58 | 79.93% ± 1.57 | 80.45% ± 1.51 |
| Llama 70B | 60.03% ± 2.33 | 71.82% ± 1.98 | 73.37% ± 1.75 | 79.26% ± 1.59 | 81.78% ± 1.45 | 84.20% ± 1.27 |
| DeepSeekV3 | 58.92% ± 2.33 | 72.55% ± 1.99 | 73.64% ± 1.76 | 79.44% ± 1.61 | 81.52% ± 1.50 | 83.68% ± 1.36 |
| GPT-4o Mini | 60.51% ± 2.26 | 72.04% ± 1.98 | 74.61% ± 1.69 | 78.25% ± 1.63 | 81.76% ± 1.43 | 83.07% ± 1.33 |
| GPT-4o | 58.06% ± 2.37 | 69.99% ± 2.12 | 73.06% ± 1.79 | 77.77% ± 1.71 | 80.69% ± 1.52 | 82.78% ± 1.42 |

| | Qwen 72B | Llama 8B | Llama 70B | DeepSeekV3 | GPT-4o Mini | GPT-4o |
|---|---|---|---|---|---|---|
| Qwen 0.5B | 60.22% ± 2.51 | 62.79% ± 2.21 | 59.91% ± 2.50 | 58.71% ± 2.55 | 60.59% ± 2.42 | 57.47% ± 2.59 |
| Qwen 1.5B | 76.11% ± 1.90 | 76.19% ± 1.69 | 75.19% ± 1.95 | 75.78% ± 1.99 | 74.21% ± 1.96 | 73.23% ± 2.15 |
| Qwen 3B | 78.69% ± 1.75 | 78.03% ± 1.59 | 77.75% ± 1.83 | 78.33% ± 1.81 | 78.80% ± 1.68 | 77.51% ± 1.87 |
| Qwen 7B | 82.57% ± 1.53 | 79.27% ± 1.59 | 81.86% ± 1.60 | 82.56% ± 1.65 | 79.25% ± 1.76 | 79.75% ± 1.85 |
| Qwen 14B | 85.50% ± 1.30 | 78.79% ± 1.60 | 83.63% ± 1.45 | 84.63% ± 1.48 | 82.42% ± 1.53 | 82.89% ± 1.64 |
| Qwen 32B | 86.49% ± 1.12 | 78.62% ± 1.55 | 85.38% ± 1.25 | 86.03% ± 1.32 | 82.60% ± 1.43 | 84.51% ± 1.45 |
| Qwen 72B | **91.35% ± 0.61** | 77.26% ± 1.72 | 85.66% ± 1.30 | 87.48% ± 1.13 | 82.10% ± 1.58 | 85.98% ± 1.40 |
| Llama 8B | 80.32% ± 1.70 | **88.70% ± 0.68** | 80.68% ± 1.68 | 80.73% ± 1.70 | 79.02% ± 1.76 | 78.50% ± 1.93 |
| Llama 70B | 86.29% ± 1.26 | 77.96% ± 1.68 | **91.05% ± 0.65** | 85.62% ± 1.41 | 82.91% ± 1.53 | 85.07% ± 1.49 |
| DeepSeekV3 | 86.36% ± 1.15 | 77.69% ± 1.70 | 84.94% ± 1.35 | **91.77% ± 0.68** | 81.58% ± 1.67 | 85.76% ± 1.44 |
| GPT-4o Mini | 84.76% ± 1.40 | 77.64% ± 1.68 | 84.43% ± 1.40 | 84.59% ± 1.51 | **89.93% ± 0.78** | 85.50% ± 1.43 |
| GPT-4o | 85.49% ± 1.31 | 75.91% ± 1.83 | 84.49% ± 1.45 | 87.05% ± 1.21 | 83.04% ± 1.48 | **90.62% ± 0.81** |

Table 16: Accuracy of proxy LIME explanations on high school psychology of MMLU datasets: each value shows how well LIME explanations generated by the model on the **left** serve as surrogates for predicting the behavior of the model on the **top**.

|            | Qwen 0.5B        | Qwen 1.5B        | Qwen 3B          | Qwen 7B          | Qwen 14B         | Qwen 32B         |
|------------|------------------|------------------|------------------|------------------|------------------|------------------|
| Qwen 0.5B  | **88.25% ± 0.64** | 75.04% ± 2.10    | 73.04% ± 2.18    | 73.87% ± 2.18    | 73.82% ± 2.35    | 72.81% ± 2.41    |
| Qwen 1.5B  | 72.43% ± 2.23    | **88.57% ± 0.68** | 79.09% ± 1.48    | 80.55% ± 1.52    | 80.12% ± 1.67    | 80.10% ± 1.70    |
| Qwen 3B    | 70.61% ± 2.28    | 79.03% ± 1.55    | **87.24% ± 0.71** | 80.59% ± 1.54    | 82.01% ± 1.47    | 81.97% ± 1.51    |
| Qwen 7B    | 70.64% ± 2.30    | 78.60% ± 1.64    | 78.69% ± 1.56    | **89.29% ± 0.67** | 83.83% ± 1.28    | 84.24% ± 1.37    |
| Qwen 14B   | 70.09% ± 2.46    | 77.96% ± 1.74    | 78.79% ± 1.58    | 82.83% ± 1.34    | **89.63% ± 0.68** | 86.68% ± 1.09    |
| Qwen 32B   | 68.77% ± 2.50    | 76.73% ± 1.79    | 77.90% ± 1.59    | 82.17% ± 1.41    | 85.24% ± 1.16    | **89.88% ± 0.68** |
| Qwen 72B   | 69.04% ± 2.52    | 76.67% ± 1.87    | 76.67% ± 1.71    | 81.19% ± 1.51    | 83.87% ± 1.34    | 85.51% ± 1.16    |
| Llama 8B   | 69.15% ± 2.43    | 77.93% ± 1.73    | 79.20% ± 1.55    | 82.30% ± 1.41    | 83.04% ± 1.41    | 83.39% ± 1.40    |
| Llama 70B  | 69.17% ± 2.59    | 76.41% ± 1.87    | 76.37% ± 1.72    | 81.32% ± 1.51    | 83.14% ± 1.40    | 85.05% ± 1.24    |
| DeepSeekV3 | 67.48% ± 2.54    | 76.95% ± 1.89    | 76.78% ± 1.74    | 81.31% ± 1.51    | 83.27% ± 1.38    | 85.16% ± 1.21    |
| GPT-4o Mini | 69.71% ± 2.45   | 76.70% ± 1.87    | 77.26% ± 1.65    | 80.63% ± 1.50    | 83.59% ± 1.32    | 84.32% ± 1.26    |
| GPT-4o     | 67.44% ± 2.63    | 75.65% ± 1.96    | 76.24% ± 1.78    | 80.41% ± 1.60    | 82.65% ± 1.45    | 84.33% ± 1.31    |

|            | Qwen 72B         | Llama 8B         | Llama 70B        | DeepSeekV3       | GPT-4o Mini      | GPT-4o           |
|------------|------------------|------------------|------------------|------------------|------------------|------------------|
| Qwen 0.5B  | 73.54% ± 2.44    | 72.37% ± 2.35    | 73.02% ± 2.53    | 71.91% ± 2.51    | 73.67% ± 2.36    | 72.08% ± 2.58    |
| Qwen 1.5B  | 80.93% ± 1.73    | 79.69% ± 1.66    | 80.31% ± 1.75    | 81.04% ± 1.79    | 79.70% ± 1.76    | 79.79% ± 1.87    |
| Qwen 3B    | 82.04% ± 1.61    | 81.16% ± 1.51    | 81.33% ± 1.68    | 82.42% ± 1.65    | 81.85% ± 1.56    | 81.78% ± 1.71    |
| Qwen 7B    | 84.59% ± 1.37    | 82.06% ± 1.46    | 84.30% ± 1.44    | 85.05% ± 1.45    | 82.25% ± 1.50    | 83.44% ± 1.56    |
| Qwen 14B   | 87.18% ± 1.17    | 81.97% ± 1.46    | 85.33% ± 1.35    | 86.72% ± 1.28    | 84.89% ± 1.29    | 85.86% ± 1.38    |
| Qwen 32B   | 87.42% ± 1.04    | 81.18% ± 1.51    | 86.17% ± 1.23    | 87.56% ± 1.12    | 84.28% ± 1.29    | 86.38% ± 1.25    |
| Qwen 72B   | **91.35% ± 0.61** | 80.19% ± 1.68    | 86.53% ± 1.23    | 88.09% ± 1.10    | 84.42% ± 1.30    | 87.40% ± 1.19    |
| Llama 8B   | 83.76% ± 1.52    | **88.70% ± 0.68** | 83.95% ± 1.49    | 84.12% ± 1.50    | 82.57% ± 1.54    | 82.98% ± 1.63    |
| Llama 70B  | 87.20% ± 1.18    | 80.96% ± 1.61    | **91.05% ± 0.65** | 87.22% ± 1.25    | 84.82% ± 1.30    | 86.94% ± 1.24    |
| DeepSeekV3 | 86.97% ± 1.10    | 80.37% ± 1.64    | 86.26% ± 1.25    | **91.77% ± 0.68** | 84.10% ± 1.39    | 87.40% ± 1.20    |
| GPT-4o Mini | 86.67% ± 1.17   | 80.70% ± 1.59    | 86.04% ± 1.22    | 86.66% ± 1.30    | **89.93% ± 0.78** | 86.88% ± 1.29    |
| GPT-4o     | 86.68% ± 1.14    | 79.34% ± 1.73    | 86.06% ± 1.25    | 87.99% ± 1.12    | 84.46% ± 1.36    | **90.62% ± 0.81** |

Table 17: Accuracy of proxy **filtered** LIME explanations on high school psychology of MMLU datasets: each value shows how well LIME explanations generated by the model on the **left** serve as surrogates for predicting the behavior of the model on the **top**.

|            | Qwen 0.5B        | Qwen 1.5B        | Qwen 3B          | Qwen 7B          | Qwen 14B         | Qwen 32B         |
|------------|------------------|------------------|------------------|------------------|------------------|------------------|
| Qwen 0.5B  | **90.36% ± 0.91** | 68.25% ± 3.34    | 63.86% ± 3.73    | 65.07% ± 3.85    | 64.21% ± 3.95    | 63.09% ± 4.10    |
| Qwen 1.5B  | 69.49% ± 3.18    | **91.99% ± 0.83** | 76.31% ± 2.98    | 73.84% ± 3.29    | 73.58% ± 3.46    | 72.11% ± 3.70    |
| Qwen 3B    | 65.82% ± 3.52    | 74.68% ± 3.05    | **93.28% ± 0.78** | 79.03% ± 2.87    | 79.62% ± 2.87    | 79.05% ± 3.08    |
| Qwen 7B    | 65.24% ± 3.67    | 72.01% ± 3.37    | 78.18% ± 2.93    | **94.01% ± 0.77** | 84.16% ± 2.38    | 83.29% ± 2.60    |
| Qwen 14B   | 64.82% ± 3.70    | 71.56% ± 3.44    | 77.93% ± 2.98    | 83.48% ± 2.36    | **94.83% ± 0.74** | 89.08% ± 1.84    |
| Qwen 32B   | 64.62% ± 3.75    | 71.06% ± 3.56    | 77.67% ± 3.03    | 82.73% ± 2.52    | 88.93% ± 1.80    | **95.39% ± 0.75** |
| Qwen 72B   | 64.12% ± 3.82    | 70.28% ± 3.66    | 77.60% ± 3.18    | 81.59% ± 2.80    | 87.11% ± 2.16    | 87.85% ± 2.24    |
| Llama 8B   | 65.05% ± 3.64    | 72.43% ± 3.34    | 79.46% ± 2.81    | 81.47% ± 2.60    | 82.76% ± 2.62    | 82.49% ± 2.75    |
| Llama 70B  | 64.27% ± 3.83    | 70.04% ± 3.65    | 77.32% ± 3.10    | 81.53% ± 2.71    | 86.75% ± 2.16    | 87.39% ± 2.27    |
| DeepSeekV3 | 63.64% ± 4.01    | 70.65% ± 3.73    | 76.50% ± 3.36    | 81.45% ± 2.83    | 86.32% ± 2.35    | 86.76% ± 2.45    |
| GPT-4o Mini | 64.55% ± 3.79   | 70.86% ± 3.61    | 77.83% ± 3.06    | 82.42% ± 2.65    | 86.04% ± 2.29    | 85.71% ± 2.56    |
| GPT-4o     | 63.23% ± 3.98    | 69.48% ± 3.79    | 75.88% ± 3.42    | 81.55% ± 2.80    | 86.39% ± 2.30    | 87.06% ± 2.40    |

|            | Qwen 72B         | Llama 8B         | Llama 70B        | DeepSeekV3       | GPT-4o Mini      | GPT-4o           |
|------------|------------------|------------------|------------------|------------------|------------------|------------------|
| Qwen 0.5B  | 62.71% ± 4.21    | 63.99% ± 3.88    | 62.61% ± 4.18    | 62.32% ± 4.38    | 63.97% ± 4.04    | 61.13% ± 4.41    |
| Qwen 1.5B  | 71.75% ± 3.84    | 73.66% ± 3.35    | 71.11% ± 3.73    | 72.88% ± 3.83    | 72.11% ± 3.70    | 70.35% ± 3.99    |
| Qwen 3B    | 79.03% ± 3.28    | 79.90% ± 2.84    | 78.50% ± 3.23    | 78.03% ± 3.49    | 79.14% ± 3.09    | 77.49% ± 3.56    |
| Qwen 7B    | 83.03% ± 2.84    | 81.11% ± 2.71    | 82.05% ± 2.85    | 82.64% ± 2.96    | 83.01% ± 2.78    | 82.70% ± 2.93    |
| Qwen 14B   | 87.92% ± 2.20    | 81.78% ± 2.72    | 87.24% ± 2.22    | 87.44% ± 2.38    | 86.08% ± 2.29    | 87.39% ± 2.36    |
| Qwen 32B   | 88.52% ± 2.23    | 81.44% ± 2.79    | 87.95% ± 2.19    | 87.76% ± 2.42    | 85.73% ± 2.45    | 88.09% ± 2.37    |
| Qwen 72B   | **96.52% ± 0.59** | 81.28% ± 3.01    | 89.10% ± 2.09    | 89.58% ± 2.24    | 86.52% ± 2.49    | 90.36% ± 2.15    |
| Llama 8B   | 82.54% ± 3.03    | **93.42% ± 0.84** | 82.66% ± 2.82    | 82.56% ± 3.08    | 82.79% ± 2.76    | 82.05% ± 3.15    |
| Llama 70B  | 89.32% ± 2.13    | 81.60% ± 2.82    | **95.32% ± 0.77** | 88.75% ± 2.26    | 85.94% ± 2.48    | 89.57% ± 2.19    |
| DeepSeekV3 | 89.23% ± 2.24    | 81.05% ± 3.03    | 87.85% ± 2.36    | **96.40% ± 0.70** | 85.64% ± 2.57    | 91.06% ± 1.98    |
| GPT-4o Mini | 87.32% ± 2.54   | 82.60% ± 2.71    | 87.26% ± 2.32    | 87.26% ± 2.43    | **95.49% ± 0.70** | 87.23% ± 2.54    |
| GPT-4o     | 90.03% ± 2.10    | 80.74% ± 3.09    | 88.63% ± 2.27    | 90.94% ± 1.97    | 86.06% ± 2.53    | **96.75% ± 0.61** |

Table 18: Accuracy of proxy LIME explanations on high school world history of MMLU datasets: each value shows how well LIME explanations generated by the model on the **left** serve as surrogates for predicting the behavior of the model on the **top**.

|  | Qwen 0.5B | Qwen 1.5B | Qwen 3B | Qwen 7B | Qwen 14B | Qwen 32B |
|---|---|---|---|---|---|---|
| Qwen 0.5B | **90.36% ± 0.91** | 76.59% ± 3.51 | 74.15% ± 3.92 | 76.08% ± 3.77 | 77.79% ± 3.81 | 78.14% ± 3.76 |
| Qwen 1.5B | 76.23% ± 3.48 | **91.99% ± 0.83** | 79.96% ± 2.99 | 80.04% ± 3.27 | 80.87% ± 3.28 | 80.63% ± 3.34 |
| Qwen 3B | 73.98% ± 3.87 | 79.24% ± 2.99 | **93.28% ± 0.78** | 80.66% ± 3.03 | 83.21% ± 2.86 | 83.12% ± 2.91 |
| Qwen 7B | 75.56% ± 3.68 | 78.94% ± 3.24 | 79.94% ± 3.03 | **94.01% ± 0.77** | 86.40% ± 2.26 | 86.03% ± 2.39 |
| Qwen 14B | 76.69% ± 3.72 | 79.12% ± 3.26 | 81.81% ± 2.87 | 85.34% ± 2.31 | **94.83% ± 0.74** | 90.40% ± 1.76 |
| Qwen 32B | 76.53% ± 3.59 | 78.90% ± 3.25 | 81.25% ± 2.93 | 84.70% ± 2.42 | 89.95% ± 1.77 | **95.39% ± 0.75** |
| Qwen 72B | 75.24% ± 3.82 | 78.44% ± 3.40 | 81.38% ± 3.06 | 84.27% ± 2.67 | 89.40% ± 1.92 | 89.90% ± 2.06 |
| Llama 8B | 75.64% ± 3.70 | 78.88% ± 3.29 | 81.76% ± 2.81 | 83.92% ± 2.59 | 85.68% ± 2.55 | 85.21% ± 2.64 |
| Llama 70B | 76.14% ± 3.90 | 78.74% ± 3.36 | 81.53% ± 2.96 | 83.56% ± 2.68 | 89.29% ± 1.96 | 89.69% ± 2.03 |
| DeepSeekV3 | 75.74% ± 3.92 | 79.76% ± 3.37 | 80.39% ± 3.29 | 83.34% ± 2.82 | 88.67% ± 2.08 | 88.98% ± 2.11 |
| GPT-4o Mini | 76.04% ± 3.63 | 78.45% ± 3.49 | 81.58% ± 2.91 | 84.88% ± 2.53 | 88.84% ± 2.03 | 88.13% ± 2.35 |
| GPT-4o | 76.52% ± 3.87 | 78.35% ± 3.43 | 79.97% ± 3.30 | 83.27% ± 2.86 | 88.40% ± 2.20 | 88.81% ± 2.23 |

|  | Qwen 72B | Llama 8B | Llama 70B | DeepSeekV3 | GPT-4o Mini | GPT-4o |
|---|---|---|---|---|---|---|
| Qwen 0.5B | 76.94% ± 3.95 | 76.53% ± 3.79 | 77.58% ± 3.98 | 77.58% ± 4.04 | 77.23% ± 3.77 | 78.31% ± 4.00 |
| Qwen 1.5B | 80.67% ± 3.47 | 80.07% ± 3.28 | 80.49% ± 3.42 | 82.30% ± 3.44 | 80.00% ± 3.54 | 80.51% ± 3.56 |
| Qwen 3B | 83.64% ± 3.08 | 82.65% ± 2.75 | 83.40% ± 2.95 | 82.79% ± 3.28 | 83.13% ± 2.93 | 82.49% ± 3.32 |
| Qwen 7B | 86.09% ± 2.63 | 84.18% ± 2.63 | 85.05% ± 2.67 | 85.67% ± 2.70 | 85.94% ± 2.54 | 85.18% ± 2.81 |
| Qwen 14B | 90.45% ± 1.86 | 85.09% ± 2.56 | 89.95% ± 1.98 | 89.98% ± 2.04 | 88.94% ± 2.05 | 89.87% ± 2.12 |
| Qwen 32B | 90.58% ± 2.03 | 84.23% ± 2.65 | 89.91% ± 2.03 | 89.82% ± 2.08 | 87.65% ± 2.35 | 89.77% ± 2.18 |
| Qwen 72B | **96.52% ± 0.59** | 85.49% ± 2.69 | 90.20% ± 2.06 | 91.99% ± 1.81 | 88.03% ± 2.38 | 92.37% ± 1.80 |
| Llama 8B | 86.96% ± 2.67 | **93.42% ± 0.84** | 87.26% ± 2.47 | 88.00% ± 2.59 | 86.85% ± 2.48 | 86.80% ± 2.69 |
| Llama 70B | 90.72% ± 2.03 | 85.99% ± 2.50 | **95.32% ± 0.77** | 90.08% ± 2.12 | 88.27% ± 2.24 | 90.49% ± 2.13 |
| DeepSeekV3 | 91.51% ± 1.89 | 85.79% ± 2.72 | 89.16% ± 2.23 | **96.40% ± 0.70** | 87.36% ± 2.46 | 92.85% ± 1.61 |
| GPT-4o Mini | 89.12% ± 2.34 | 86.28% ± 2.45 | 89.12% ± 2.21 | 88.85% ± 2.35 | **95.49% ± 0.70** | 89.63% ± 2.33 |
| GPT-4o | 91.73% ± 1.90 | 84.95% ± 2.73 | 89.56% ± 2.22 | 92.57% ± 1.65 | 88.03% ± 2.43 | **96.75% ± 0.61** |

Table 19: Accuracy of proxy **filtered** LIME explanations on high school world history of MMLU datasets: each value shows how well LIME explanations generated by the model on the **left** serve as surrogates for predicting the behavior of the model on the **top**.

|  | Qwen 0.5B | Qwen 1.5B | Qwen 3B | Qwen 7B | Qwen 14B | Qwen 32B |
|---|---|---|---|---|---|---|
| Qwen 0.5B | **79.57% ± 2.29** | 57.58% ± 3.83 | 55.11% ± 4.00 | 52.69% ± 3.94 | 52.11% ± 3.89 | 51.86% ± 4.08 |
| Qwen 1.5B | 56.67% ± 3.75 | **76.57% ± 2.48** | 65.22% ± 3.40 | 61.13% ± 3.58 | 61.17% ± 3.44 | 60.88% ± 3.62 |
| Qwen 3B | 53.17% ± 4.02 | 63.62% ± 3.28 | **76.13% ± 2.43** | 64.01% ± 3.36 | 65.89% ± 3.19 | 65.00% ± 3.51 |
| Qwen 7B | 50.78% ± 4.08 | 59.90% ± 3.71 | 64.21% ± 3.38 | **75.91% ± 2.62** | 68.60% ± 2.87 | 65.86% ± 3.49 |
| Qwen 14B | 52.47% ± 3.86 | 60.29% ± 3.51 | 62.05% ± 3.49 | 66.96% ± 3.20 | **72.97% ± 2.73** | 69.70% ± 3.09 |
| Qwen 32B | 51.49% ± 4.08 | 59.04% ± 3.87 | 61.37% ± 3.68 | 66.05% ± 3.30 | 67.24% ± 3.25 | **78.10% ± 2.47** |
| Qwen 72B | 51.61% ± 4.14 | 59.96% ± 3.83 | 62.77% ± 3.57 | 67.94% ± 3.28 | 67.42% ± 3.23 | 71.90% ± 3.21 |
| Llama 8B | 50.51% ± 3.97 | 58.38% ± 3.80 | 62.09% ± 3.44 | 64.93% ± 3.28 | 63.94% ± 3.23 | 63.82% ± 3.58 |
| Llama 70B | 52.08% ± 3.96 | 59.40% ± 3.81 | 62.38% ± 3.68 | 65.09% ± 3.41 | 66.48% ± 3.31 | 71.06% ± 2.96 |
| DeepSeekV3 | 52.15% ± 4.19 | 59.74% ± 3.90 | 64.14% ± 3.43 | 66.82% ± 3.33 | 67.48% ± 3.20 | 72.30% ± 3.19 |
| GPT-4o Mini | 51.72% ± 4.16 | 57.42% ± 3.92 | 62.46% ± 3.57 | 64.06% ± 3.52 | 67.18% ± 3.27 | 67.59% ± 3.40 |
| GPT-4o | 52.27% ± 4.18 | 53.93% ± 3.98 | 59.94% ± 3.79 | 62.91% ± 3.62 | 65.29% ± 3.46 | 68.00% ± 3.54 |

|  | Qwen 72B | Llama 8B | Llama 70B | DeepSeekV3 | GPT-4o Mini | GPT-4o |
|---|---|---|---|---|---|---|
| Qwen 0.5B | 52.37% ± 4.14 | 50.76% ± 3.66 | 52.13% ± 3.91 | 49.19% ± 4.10 | 53.86% ± 3.98 | 51.17% ± 4.32 |
| Qwen 1.5B | 60.18% ± 3.82 | 59.53% ± 3.49 | 60.26% ± 3.66 | 59.45% ± 3.63 | 61.10% ± 3.63 | 55.91% ± 4.23 |
| Qwen 3B | 62.91% ± 3.69 | 61.13% ± 3.42 | 64.79% ± 3.36 | 62.55% ± 3.59 | 62.76% ± 3.48 | 58.50% ± 4.30 |
| Qwen 7B | 67.59% ± 3.45 | 62.75% ± 3.30 | 66.82% ± 3.26 | 67.06% ± 3.32 | 62.73% ± 3.58 | 55.86% ± 4.45 |
| Qwen 14B | 69.30% ± 3.31 | 60.49% ± 3.47 | 67.95% ± 3.17 | 68.56% ± 3.20 | 65.43% ± 3.31 | 60.26% ± 4.17 |
| Qwen 32B | 71.60% ± 3.26 | 61.20% ± 3.37 | 69.65% ± 3.18 | 72.82% ± 2.94 | 66.38% ± 3.51 | 63.95% ± 4.15 |
| Qwen 72B | **79.57% ± 2.54** | 60.77% ± 3.58 | 70.85% ± 3.22 | 72.52% ± 3.09 | 67.61% ± 3.48 | 64.93% ± 4.24 |
| Llama 8B | 62.59% ± 3.74 | **75.51% ± 2.39** | 66.46% ± 3.34 | 64.92% ± 3.53 | 62.24% ± 3.64 | 54.18% ± 4.47 |
| Llama 70B | 71.09% ± 3.28 | 62.62% ± 3.44 | **77.52% ± 2.58** | 71.37% ± 3.03 | 66.51% ± 3.57 | 63.95% ± 4.28 |
| DeepSeekV3 | 70.80% ± 3.51 | 61.44% ± 3.55 | 69.22% ± 3.35 | **78.82% ± 2.52** | 67.41% ± 3.42 | 65.96% ± 4.14 |
| GPT-4o Mini | 70.99% ± 3.38 | 57.86% ± 3.68 | 67.77% ± 3.43 | 70.57% ± 3.27 | **70.62% ± 3.26** | 68.27% ± 3.72 |
| GPT-4o | 70.64% ± 3.51 | 57.77% ± 3.61 | 67.25% ± 3.43 | 72.61% ± 3.18 | 66.68% ± 3.46 | **71.38% ± 3.50** |

Table 20: Accuracy of proxy Kernel SHAP explanations on high school chemistriy of MMLU datasets: each value shows how well Kernel SHAP explanations generated by the model on the **left** serve as surrogates for predicting the behavior of the model on the **top**.

|  | Qwen 0.5B | Qwen 1.5B | Qwen 3B | Qwen 7B | Qwen 14B | Qwen 32B |
|---|---|---|---|---|---|---|
| Qwen 0.5B | **79.57%** ± **2.29** | 67.42% ± 4.66 | 68.75% ± 5.00 | 64.16% ± 5.44 | 67.35% ± 6.09 | 72.14% ± 5.57 |
| Qwen 1.5B | 70.51% ± 4.43 | **76.57%** ± **2.48** | 72.57% ± 3.55 | 70.46% ± 4.17 | 69.40% ± 4.06 | 71.97% ± 4.24 |
| Qwen 3B | 70.46% ± 5.07 | 69.16% ± 4.09 | **76.13%** ± **2.43** | 70.79% ± 3.82 | 72.63% ± 3.75 | 74.54% ± 3.63 |
| Qwen 7B | 65.90% ± 5.45 | 67.53% ± 4.51 | 70.18% ± 3.63 | **75.91%** ± **2.62** | 73.07% ± 3.32 | 74.09% ± 3.54 |
| Qwen 14B | 70.50% ± 5.14 | 69.12% ± 3.96 | 68.63% ± 4.02 | 72.42% ± 3.56 | **72.97%** ± **2.73** | 73.58% ± 3.15 |
| Qwen 32B | 69.56% ± 5.99 | 68.68% ± 4.63 | 68.40% ± 4.41 | 72.28% ± 3.74 | 71.45% ± 3.55 | **78.10%** ± **2.47** |
| Qwen 72B | 69.70% ± 5.71 | 70.44% ± 4.73 | 70.33% ± 4.11 | 74.52% ± 3.21 | 73.21% ± 3.53 | 75.92% ± 3.11 |
| Llama 8B | 68.94% ± 5.82 | 67.18% ± 4.75 | 68.81% ± 4.26 | 71.85% ± 3.63 | 71.10% ± 3.61 | 73.71% ± 3.50 |
| Llama 70B | 71.21% ± 5.34 | 69.64% ± 4.36 | 71.16% ± 4.09 | 72.81% ± 3.65 | 71.75% ± 3.61 | 75.07% ± 2.93 |
| DeepSeekV3 | 71.98% ± 5.65 | 69.87% ± 4.53 | 72.20% ± 3.88 | 73.35% ± 3.59 | 73.42% ± 3.57 | 75.62% ± 3.13 |
| GPT-4o Mini | 71.25% ± 5.45 | 67.38% ± 4.84 | 68.52% ± 3.98 | 71.75% ± 4.04 | 71.94% ± 3.97 | 74.18% ± 3.69 |
| GPT-4o | 70.86% ± 5.54 | 61.56% ± 5.10 | 67.62% ± 4.67 | 70.94% ± 4.29 | 70.84% ± 4.16 | 74.26% ± 3.76 |
|  | **Qwen 72B** | **Llama 8B** | **Llama 70B** | **DeepSeekV3** | **GPT-4o Mini** | **GPT-4o** |
| Qwen 0.5B | 71.04% ± 5.32 | 65.76% ± 5.67 | 71.43% ± 5.32 | 67.91% ± 6.00 | 67.45% ± 5.58 | 73.38% ± 5.06 |
| Qwen 1.5B | 73.40% ± 4.23 | 70.67% ± 4.05 | 72.48% ± 4.16 | 71.46% ± 4.12 | 70.20% ± 4.24 | 70.77% ± 4.27 |
| Qwen 3B | 73.17% ± 3.83 | 69.04% ± 3.95 | 72.75% ± 3.81 | 72.87% ± 3.98 | 68.48% ± 3.89 | 72.95% ± 3.87 |
| Qwen 7B | 74.22% ± 3.52 | 69.55% ± 3.78 | 72.96% ± 3.69 | 73.94% ± 3.69 | 69.22% ± 4.08 | 72.63% ± 3.72 |
| Qwen 14B | 74.86% ± 3.27 | 69.10% ± 3.68 | 73.18% ± 3.38 | 74.73% ± 3.37 | 71.11% ± 3.62 | 72.94% ± 3.48 |
| Qwen 32B | 76.50% ± 3.07 | 69.62% ± 3.64 | 73.22% ± 3.28 | 76.26% ± 2.95 | 72.94% ± 3.44 | 75.47% ± 3.15 |
| Qwen 72B | **79.57%** ± **2.54** | 69.68% ± 3.66 | 74.78% ± 3.36 | 76.62% ± 3.06 | 71.35% ± 3.71 | 76.18% ± 3.36 |
| Llama 8B | 73.27% ± 3.69 | **75.51%** ± **2.39** | 74.03% ± 3.56 | 74.51% ± 3.55 | 70.98% ± 4.02 | 71.60% ± 4.17 |
| Llama 70B | 76.37% ± 3.22 | 71.22% ± 3.52 | **77.52%** ± **2.58** | 75.51% ± 3.05 | 72.90% ± 3.51 | 76.19% ± 3.28 |
| DeepSeekV3 | 75.76% ± 3.54 | 70.13% ± 3.62 | 73.24% ± 3.45 | **78.82%** ± **2.52** | 73.35% ± 3.62 | 75.97% ± 3.22 |
| GPT-4o Mini | 76.20% ± 3.44 | 68.34% ± 4.09 | 73.39% ± 3.71 | 76.63% ± 3.48 | **70.62%** ± **3.26** | 76.11% ± 3.36 |
| GPT-4o | 76.07% ± 3.70 | 68.22% ± 4.21 | 72.60% ± 3.84 | 76.87% ± 3.42 | 67.41% ± 4.05 | **71.38%** ± **3.50** |

Table 21: Accuracy of **filtered** proxy Kernel SHAP explanations on high school chemistriy of MMLU datasets: each value shows how well Kernel SHAP explanations generated by the model on the **left** serve as surrogates for predicting the behavior of the model on the **top**.

|  | Qwen 0.5B | Qwen 1.5B | Qwen 3B | Qwen 7B | Qwen 14B | Qwen 32B |
|---|---|---|---|---|---|---|
| Qwen 0.5B | **78.94%** ± **3.20** | 62.47% ± 5.17 | 61.97% ± 4.79 | 60.71% ± 5.16 | 57.75% ± 5.15 | 59.20% ± 5.11 |
| Qwen 1.5B | 61.14% ± 5.59 | **75.69%** ± **3.63** | 67.03% ± 4.79 | 69.18% ± 4.81 | 66.45% ± 4.79 | 64.11% ± 5.07 |
| Qwen 3B | 61.63% ± 5.22 | 66.26% ± 4.49 | **72.73%** ± **3.81** | 70.34% ± 4.33 | 69.87% ± 4.28 | 70.31% ± 4.45 |
| Qwen 7B | 59.00% ± 5.40 | 64.09% ± 4.69 | 66.83% ± 4.46 | **73.20%** ± **3.74** | 70.19% ± 4.11 | 71.18% ± 3.85 |
| Qwen 14B | 59.04% ± 5.47 | 64.16% ± 4.84 | 69.60% ± 4.20 | 73.69% ± 3.97 | **74.34%** ± **4.09** | 75.84% ± 3.90 |
| Qwen 32B | 52.86% ± 3.93 | 57.94% ± 3.56 | 59.09% ± 3.25 | 61.35% ± 3.16 | 62.02% ± 3.14 | **62.40%** ± **3.19** |
| Qwen 72B | 58.06% ± 5.49 | 62.79% ± 5.17 | 67.77% ± 4.36 | 72.77% ± 3.81 | 71.57% ± 4.26 | 75.13% ± 3.71 |
| Llama 8B | 58.28% ± 5.67 | 61.59% ± 5.23 | 65.50% ± 4.66 | 69.26% ± 4.88 | 67.55% ± 4.79 | 66.63% ± 4.97 |
| Llama 70B | 59.01% ± 5.86 | 63.51% ± 5.09 | 66.50% ± 4.82 | 72.90% ± 4.03 | 70.47% ± 4.80 | 74.90% ± 3.85 |
| DeepSeekV3 | 57.81% ± 5.77 | 63.22% ± 5.06 | 65.30% ± 4.63 | 71.31% ± 4.37 | 70.73% ± 4.64 | 72.04% ± 4.52 |
| GPT-4o Mini | 61.37% ± 5.48 | 64.37% ± 5.01 | 65.37% ± 4.69 | 69.66% ± 4.45 | 69.46% ± 4.59 | 71.12% ± 4.35 |
| GPT-4o | 57.59% ± 5.84 | 59.38% ± 5.61 | 62.93% ± 5.11 | 66.82% ± 4.81 | 66.55% ± 5.28 | 69.07% ± 4.86 |
|  | **Qwen 72B** | **Llama 8B** | **Llama 70B** | **DeepSeekV3** | **GPT-4o Mini** | **GPT-4o** |
| Qwen 0.5B | 58.02% ± 5.28 | 61.61% ± 4.81 | 59.14% ± 5.15 | 59.99% ± 5.43 | 58.67% ± 5.54 | 56.69% ± 5.79 |
| Qwen 1.5B | 64.27% ± 5.10 | 65.30% ± 4.85 | 66.07% ± 5.04 | 64.62% ± 5.30 | 63.27% ± 5.14 | 62.28% ± 5.79 |
| Qwen 3B | 69.72% ± 4.47 | 66.63% ± 4.25 | 70.84% ± 4.17 | 68.22% ± 4.68 | 68.25% ± 4.33 | 66.74% ± 4.94 |
| Qwen 7B | 70.50% ± 4.10 | 66.03% ± 4.51 | 70.67% ± 3.91 | 69.92% ± 4.22 | 67.35% ± 4.24 | 67.61% ± 4.51 |
| Qwen 14B | 74.06% ± 4.09 | 68.61% ± 4.54 | 73.41% ± 4.03 | 72.76% ± 4.58 | 70.20% ± 4.68 | 70.67% ± 5.02 |
| Qwen 32B | 61.44% ± 3.21 | 59.41% ± 3.39 | 59.63% ± 3.34 | 60.41% ± 3.30 | 58.72% ± 3.43 | 59.49% ± 3.61 |
| Qwen 72B | **75.60%** ± **3.57** | 66.92% ± 4.47 | 74.79% ± 3.85 | 72.75% ± 4.19 | 67.75% ± 4.81 | 71.22% ± 4.47 |
| Llama 8B | 67.39% ± 4.82 | **70.03%** ± **4.27** | 68.70% ± 4.71 | 66.92% ± 5.15 | 65.75% ± 5.04 | 65.16% ± 5.42 |
| Llama 70B | 72.76% ± 4.30 | 67.10% ± 4.77 | **76.91%** ± **3.65** | 71.63% ± 4.53 | 66.94% ± 4.91 | 67.30% ± 5.31 |
| DeepSeekV3 | 70.78% ± 4.60 | 66.43% ± 4.68 | 72.68% ± 4.19 | **75.02%** ± **4.14** | 67.05% ± 5.06 | 68.70% ± 4.94 |
| GPT-4o Mini | 70.65% ± 4.48 | 63.73% ± 4.97 | 70.58% ± 4.52 | 71.24% ± 4.72 | **70.29%** ± **4.94** | 73.27% ± 4.72 |
| GPT-4o | 68.38% ± 4.88 | 62.70% ± 5.13 | 68.84% ± 4.76 | 69.57% ± 4.81 | 67.85% ± 5.19 | **68.19%** ± **5.62** |

Table 22: Accuracy of proxy Kernel SHAP explanations on high school computer science of MMLU datasets: each value shows how well Kernel SHAP explanations generated by the model on the **left** serve as surrogates for predicting the behavior of the model on the **top**.

|  | Qwen 0.5B | Qwen 1.5B | Qwen 3B | Qwen 7B | Qwen 14B | Qwen 32B |
|---|---|---|---|---|---|---|
| Qwen 0.5B | **78.94%** ± **3.20** | 73.30% ± 5.79 | 71.40% ± 5.84 | 69.41% ± 7.09 | 68.97% ± 7.31 | 55.29% ± 7.38 |
| Qwen 1.5B | 71.83% ± 6.37 | **75.69%** ± **3.63** | 72.60% ± 5.37 | 75.67% ± 5.13 | 73.31% ± 5.45 | 64.04% ± 8.61 |
| Qwen 3B | 71.80% ± 5.76 | 69.96% ± 4.74 | **72.73%** ± **3.81** | 71.95% ± 5.08 | 73.40% ± 5.04 | 64.50% ± 7.61 |
| Qwen 7B | 69.11% ± 6.31 | 70.92% ± 5.00 | 70.42% ± 5.09 | **73.20%** ± **3.74** | 71.46% ± 4.58 | 69.11% ± 6.32 |
| Qwen 14B | 72.27% ± 6.55 | 72.50% ± 4.98 | 74.73% ± 4.58 | 75.39% ± 4.35 | **74.34%** ± **4.09** | 70.47% ± 7.47 |
| Qwen 32B | 51.87% ± 5.21 | 56.86% ± 5.96 | 55.80% ± 5.37 | 62.33% ± 5.40 | 62.89% ± 5.26 | **62.40%** ± **3.19** |
| Qwen 72B | 69.31% ± 6.79 | 71.64% ± 4.91 | 73.27% ± 4.65 | 73.26% ± 4.22 | 72.96% ± 4.36 | 70.74% ± 7.97 |
| Llama 8B | 70.88% ± 7.60 | 68.47% ± 5.71 | 69.63% ± 5.30 | 71.86% ± 5.62 | 71.08% ± 5.52 | 70.90% ± 9.64 |
| Llama 70B | 72.58% ± 7.02 | 72.12% ± 5.19 | 71.33% ± 5.39 | 73.48% ± 4.43 | 71.71% ± 5.06 | 70.92% ± 7.83 |
| DeepSeekV3 | 70.49% ± 6.87 | 70.88% ± 5.81 | 71.81% ± 5.28 | 72.24% ± 4.78 | 72.83% ± 4.88 | 64.75% ± 10.47 |
| GPT-4o Mini | 72.76% ± 6.06 | 70.58% ± 5.45 | 67.45% ± 5.54 | 68.70% ± 4.94 | 70.37% ± 5.31 | 65.80% ± 8.57 |
| GPT-4o | 66.78% ± 7.81 | 67.19% ± 6.41 | 67.48% ± 5.96 | 66.75% ± 5.38 | 68.45% ± 5.80 | 63.02% ± 10.25 |

|  | Qwen 72B | Llama 8B | Llama 70B | DeepSeekV3 | GPT-4o Mini | GPT-4o |
|---|---|---|---|---|---|---|
| Qwen 0.5B | 71.58% ± 7.33 | 74.66% ± 5.72 | 73.57% ± 6.62 | 72.41% ± 6.97 | 70.12% ± 7.96 | 69.07% ± 8.17 |
| Qwen 1.5B | 74.25% ± 5.43 | 70.00% ± 5.28 | 76.48% ± 4.87 | 74.20% ± 5.79 | 70.73% ± 5.87 | 71.90% ± 5.99 |
| Qwen 3B | 75.19% ± 4.98 | 68.55% ± 4.55 | 75.98% ± 4.51 | 72.74% ± 5.35 | 70.15% ± 5.05 | 70.90% ± 5.56 |
| Qwen 7B | 72.87% ± 4.21 | 68.78% ± 4.96 | 73.73% ± 3.91 | 70.59% ± 4.58 | 69.56% ± 4.42 | 69.18% ± 4.71 |
| Qwen 14B | 76.00% ± 4.26 | 72.38% ± 4.76 | 75.07% ± 4.29 | 74.14% ± 4.85 | 72.73% ± 5.03 | 72.81% ± 5.28 |
| Qwen 32B | 57.79% ± 5.32 | 58.29% ± 6.01 | 56.93% ± 5.67 | 56.43% ± 5.26 | 54.72% ± 5.83 | 51.39% ± 6.52 |
| Qwen 72B | **75.60%** ± **3.57** | 70.47% ± 4.84 | 76.41% ± 3.73 | 72.85% ± 4.30 | 70.26% ± 5.10 | 73.12% ± 4.38 |
| Llama 8B | 73.38% ± 5.18 | **70.03%** ± **4.27** | 74.60% ± 4.82 | 73.29% ± 5.45 | 71.20% ± 5.49 | 72.36% ± 5.78 |
| Llama 70B | 74.03% ± 4.32 | 70.60% ± 5.08 | **76.91%** ± **3.65** | 72.43% ± 4.70 | 69.47% ± 5.37 | 69.39% ± 5.41 |
| DeepSeekV3 | 73.10% ± 4.47 | 70.67% ± 4.93 | 74.34% ± 4.31 | **75.02%** ± **4.14** | 69.91% ± 5.42 | 70.21% ± 5.05 |
| GPT-4o Mini | 72.01% ± 4.89 | 66.35% ± 5.47 | 72.84% ± 4.79 | 72.13% ± 5.04 | **70.29%** ± **4.94** | 73.04% ± 5.12 |
| GPT-4o | 70.42% ± 5.11 | 64.97% ± 6.24 | 71.37% ± 4.94 | 69.83% ± 5.19 | 69.41% ± 5.72 | **68.19%** ± **5.62** |

Table 23: Accuracy of **filtered** proxy Kernel SHAP explanations on high school computer science of MMLU datasets: each value shows how well Kernel SHAP explanations generated by the model on the **left** serve as surrogates for predicting the behavior of the model on the **top**.

|  | Qwen 0.5B | Qwen 1.5B | Qwen 3B | Qwen 7B | Qwen 14B | Qwen 32B |
|---|---|---|---|---|---|---|
| Qwen 0.5B | **80.88%** ± **1.98** | 61.67% ± 3.43 | 57.45% ± 3.41 | 56.32% ± 3.82 | 55.58% ± 3.69 | 53.73% ± 3.81 |
| Qwen 1.5B | 59.47% ± 3.52 | **78.61%** ± **2.28** | 67.09% ± 3.10 | 67.26% ± 3.49 | 66.60% ± 3.40 | 65.98% ± 3.54 |
| Qwen 3B | 55.26% ± 3.61 | 66.02% ± 3.26 | **78.10%** ± **2.23** | 71.28% ± 3.05 | 70.98% ± 2.92 | 69.09% ± 3.14 |
| Qwen 7B | 55.17% ± 3.72 | 64.23% ± 3.41 | 68.50% ± 3.05 | **81.79%** ± **2.08** | 73.68% ± 2.80 | 76.07% ± 2.62 |
| Qwen 14B | 54.66% ± 3.65 | 62.56% ± 3.40 | 67.67% ± 3.11 | 74.66% ± 2.69 | **78.87%** ± **2.32** | 76.76% ± 2.54 |
| Qwen 32B | 54.25% ± 3.72 | 62.40% ± 3.54 | 67.24% ± 3.11 | 74.79% ± 2.79 | 75.49% ± 2.60 | **81.67%** ± **2.20** |
| Qwen 72B | 53.16% ± 3.66 | 62.17% ± 3.39 | 66.47% ± 3.11 | 74.49% ± 2.73 | 75.05% ± 2.64 | 78.53% ± 2.39 |
| Llama 8B | 57.76% ± 3.53 | 65.09% ± 3.13 | 67.47% ± 3.08 | 70.17% ± 3.25 | 70.40% ± 3.11 | 69.81% ± 3.24 |
| Llama 70B | 54.42% ± 3.74 | 63.87% ± 3.35 | 67.13% ± 3.13 | 74.33% ± 2.80 | 74.19% ± 2.84 | 75.99% ± 2.65 |
| DeepSeekV3 | 52.79% ± 3.83 | 60.88% ± 3.69 | 65.88% ± 3.38 | 73.14% ± 3.13 | 75.08% ± 2.89 | 77.05% ± 2.60 |
| GPT-4o Mini | 53.87% ± 3.73 | 63.73% ± 3.39 | 66.83% ± 3.12 | 73.03% ± 2.92 | 74.01% ± 2.84 | 76.38% ± 2.66 |
| GPT-4o | 54.03% ± 3.75 | 61.93% ± 3.45 | 65.49% ± 3.25 | 74.17% ± 2.77 | 74.12% ± 2.84 | 76.34% ± 2.73 |

|  | Qwen 72B | Llama 8B | Llama 70B | DeepSeekV3 | GPT-4o Mini | GPT-4o |
|---|---|---|---|---|---|---|
| Qwen 0.5B | 53.40% ± 4.00 | 58.72% ± 3.55 | 55.39% ± 3.84 | 53.12% ± 4.06 | 55.12% ± 3.69 | 52.46% ± 4.02 |
| Qwen 1.5B | 66.36% ± 3.71 | 68.09% ± 3.14 | 68.17% ± 3.39 | 66.08% ± 3.75 | 66.93% ± 3.39 | 65.83% ± 3.65 |
| Qwen 3B | 68.05% ± 3.31 | 67.25% ± 3.17 | 69.07% ± 3.11 | 67.31% ± 3.44 | 68.98% ± 3.05 | 66.96% ± 3.34 |
| Qwen 7B | 74.55% ± 2.79 | 68.18% ± 3.26 | 73.19% ± 2.83 | 75.48% ± 2.80 | 72.01% ± 2.88 | 74.11% ± 2.79 |
| Qwen 14B | 75.63% ± 2.75 | 66.99% ± 3.29 | 74.03% ± 2.78 | 76.79% ± 2.74 | 74.61% ± 2.74 | 75.21% ± 2.78 |
| Qwen 32B | 77.35% ± 2.73 | 67.79% ± 3.21 | 75.45% ± 2.78 | 79.40% ± 2.52 | 74.20% ± 2.80 | 76.72% ± 2.74 |
| Qwen 72B | **83.40%** ± **2.03** | 66.96% ± 3.16 | 76.92% ± 2.69 | 80.37% ± 2.41 | 74.94% ± 2.78 | 79.23% ± 2.60 |
| Llama 8B | 69.87% ± 3.30 | **79.76%** ± **2.22** | 71.90% ± 3.08 | 70.37% ± 3.42 | 69.59% ± 3.23 | 70.53% ± 3.33 |
| Llama 70B | 78.62% ± 2.61 | 69.43% ± 3.08 | **82.66%** ± **2.07** | 79.18% ± 2.59 | 75.73% ± 2.73 | 79.34% ± 2.41 |
| DeepSeekV3 | 79.11% ± 2.66 | 66.97% ± 3.38 | 75.74% ± 2.83 | **83.59%** ± **2.14** | 73.88% ± 3.01 | 77.08% ± 2.83 |
| GPT-4o Mini | 77.49% ± 2.75 | 67.53% ± 3.24 | 76.78% ± 2.75 | 77.97% ± 2.73 | **79.43%** ± **2.52** | 79.36% ± 2.53 |
| GPT-4o | 78.36% ± 2.74 | 67.54% ± 3.14 | 76.75% ± 2.66 | 78.82% ± 2.70 | 75.37% ± 2.78 | **82.00%** ± **2.28** |

Table 24: Accuracy of proxy Kernel SHAP explanations on high school microeconomics of MMLU datasets: each value shows how well Kernel SHAP explanations generated by the model on the **left** serve as surrogates for predicting the behavior of the model on the **top**.

| | Qwen 0.5B | Qwen 1.5B | Qwen 3B | Qwen 7B | Qwen 14B | Qwen 32B |
|---|---|---|---|---|---|---|
| Qwen 0.5B | **80.88%** ± **1.98** | 71.51% ± 3.83 | 70.03% ± 3.99 | 76.19% ± 3.64 | 72.03% ± 4.20 | 73.52% ± 3.93 |
| Qwen 1.5B | 74.25% ± 3.36 | **78.61%** ± **2.28** | 73.88% ± 3.15 | 77.99% ± 2.82 | 76.05% ± 2.98 | 76.95% ± 2.92 |
| Qwen 3B | 71.52% ± 3.73 | 73.73% ± 2.98 | **78.10%** ± **2.23** | 76.84% ± 2.73 | 76.58% ± 2.71 | 76.12% ± 2.89 |
| Qwen 7B | 73.92% ± 3.80 | 74.14% ± 2.88 | 73.50% ± 3.05 | **81.79%** ± **2.08** | 75.55% ± 2.89 | 78.63% ± 2.53 |
| Qwen 14B | 71.25% ± 4.05 | 71.35% ± 3.12 | 73.77% ± 3.05 | 77.18% ± 2.59 | **78.87%** ± **2.32** | 78.78% ± 2.49 |
| Qwen 32B | 72.83% ± 3.91 | 71.94% ± 3.23 | 73.06% ± 3.15 | 77.82% ± 2.60 | 76.94% ± 2.69 | **81.67%** ± **2.20** |
| Qwen 72B | 69.76% ± 4.10 | 71.24% ± 3.30 | 71.84% ± 3.19 | 76.97% ± 2.63 | 76.58% ± 2.63 | 79.73% ± 2.36 |
| Llama 8B | 73.29% ± 3.71 | 70.83% ± 3.17 | 73.58% ± 3.25 | 77.27% ± 2.80 | 76.30% ± 2.86 | 77.08% ± 2.83 |
| Llama 70B | 72.80% ± 4.00 | 72.09% ± 3.18 | 72.05% ± 3.20 | 77.04% ± 2.60 | 76.71% ± 2.69 | 77.80% ± 2.54 |
| DeepSeekV3 | 72.04% ± 4.14 | 71.02% ± 3.52 | 72.49% ± 3.37 | 76.66% ± 2.95 | 77.72% ± 2.72 | 78.28% ± 2.62 |
| GPT-4o Mini | 71.23% ± 4.08 | 72.52% ± 3.26 | 71.57% ± 3.24 | 76.25% ± 2.72 | 76.56% ± 2.78 | 78.77% ± 2.53 |
| GPT-4o | 71.55% ± 4.17 | 70.04% ± 3.38 | 70.92% ± 3.39 | 76.52% ± 2.80 | 76.10% ± 2.79 | 78.07% ± 2.61 |

| | Qwen 72B | Llama 8B | Llama 70B | DeepSeekV3 | GPT-4o Mini | GPT-4o |
|---|---|---|---|---|---|---|
| Qwen 0.5B | 74.58% ± 3.94 | 71.47% ± 3.96 | 74.22% ± 3.98 | 75.53% ± 4.09 | 70.43% ± 4.34 | 74.36% ± 4.17 |
| Qwen 1.5B | 78.92% ± 2.95 | 74.80% ± 2.91 | 78.25% ± 2.90 | 79.15% ± 2.79 | 76.23% ± 3.13 | 77.51% ± 2.93 |
| Qwen 3B | 76.03% ± 2.77 | 75.34% ± 2.82 | 75.43% ± 2.83 | 76.00% ± 2.93 | 73.94% ± 2.94 | 74.93% ± 2.87 |
| Qwen 7B | 77.57% ± 2.55 | 74.54% ± 3.00 | 76.10% ± 2.73 | 78.93% ± 2.54 | 74.64% ± 2.76 | 77.81% ± 2.50 |
| Qwen 14B | 78.32% ± 2.48 | 73.55% ± 3.06 | 76.81% ± 2.64 | 79.37% ± 2.53 | 76.87% ± 2.63 | 78.06% ± 2.50 |
| Qwen 32B | 79.46% ± 2.52 | 73.53% ± 3.12 | 77.78% ± 2.62 | 80.74% ± 2.46 | 76.27% ± 2.80 | 78.95% ± 2.50 |
| Qwen 72B | **83.40%** ± **2.03** | 72.58% ± 3.05 | 79.72% ± 2.43 | 81.16% ± 2.36 | 77.04% ± 2.60 | 81.20% ± 2.28 |
| Llama 8B | 78.18% ± 2.73 | **79.76%** ± **2.22** | 79.26% ± 2.65 | 79.66% ± 2.78 | 76.04% ± 3.07 | 78.93% ± 2.73 |
| Llama 70B | 81.90% ± 2.15 | 75.09% ± 2.88 | **82.66%** ± **2.07** | 81.63% ± 2.29 | 77.56% ± 2.67 | 81.42% ± 2.17 |
| DeepSeekV3 | 80.79% ± 2.46 | 73.96% ± 3.22 | 78.81% ± 2.57 | **83.59%** ± **2.14** | 76.16% ± 2.89 | 78.81% ± 2.57 |
| GPT-4o Mini | 80.57% ± 2.39 | 74.43% ± 3.02 | 79.33% ± 2.52 | 80.78% ± 2.41 | **79.43%** ± **2.52** | 81.61% ± 2.29 |
| GPT-4o | 80.38% ± 2.54 | 73.15% ± 3.07 | 78.49% ± 2.60 | 79.92% ± 2.59 | 76.48% ± 2.83 | **82.00%** ± **2.28** |

Table 25: Accuracy of **filtered** proxy Kernel SHAP explanations on high school microeconomics of MMLU datasets: each value shows how well Kernel SHAP explanations generated by the model on the **left** serve as surrogates for predicting the behavior of the model on the **top**.

| | Qwen 0.5B | Qwen 1.5B | Qwen 3B | Qwen 7B | Qwen 14B | Qwen 32B |
|---|---|---|---|---|---|---|
| Qwen 0.5B | **79.08%** ± **1.31** | 65.63% ± 2.05 | 63.79% ± 1.97 | 63.02% ± 2.13 | 62.30% ± 2.21 | 61.41% ± 2.22 |
| Qwen 1.5B | 63.83% ± 2.05 | **79.61%** ± **1.33** | 72.62% ± 1.61 | 74.60% ± 1.57 | 73.54% ± 1.69 | 73.37% ± 1.72 |
| Qwen 3B | 59.66% ± 2.05 | 69.23% ± 1.78 | **75.48%** ± **1.39** | 71.30% ± 1.65 | 71.97% ± 1.67 | 71.63% ± 1.67 |
| Qwen 7B | 60.72% ± 2.05 | 69.35% ± 1.68 | 69.59% ± 1.62 | **77.64%** ± **1.38** | 74.26% ± 1.56 | 74.20% ± 1.57 |
| Qwen 14B | 60.05% ± 2.02 | 67.38% ± 1.77 | 69.23% ± 1.61 | 71.54% ± 1.63 | **76.59%** ± **1.47** | 74.28% ± 1.56 |
| Qwen 32B | 60.12% ± 2.11 | 68.44% ± 1.87 | 70.53% ± 1.68 | 74.64% ± 1.63 | 77.50% ± 1.52 | **80.23%** ± **1.32** |
| Qwen 72B | 59.83% ± 2.10 | 68.97% ± 1.82 | 70.05% ± 1.65 | 74.23% ± 1.55 | 76.04% ± 1.51 | 76.85% ± 1.44 |
| Llama 8B | 60.67% ± 2.12 | 70.39% ± 1.84 | 71.71% ± 1.66 | 74.68% ± 1.68 | 75.20% ± 1.66 | 76.18% ± 1.54 |
| Llama 70B | 59.80% ± 2.24 | 68.78% ± 1.98 | 70.36% ± 1.77 | 74.87% ± 1.68 | 77.40% ± 1.59 | 78.88% ± 1.45 |
| DeepSeekV3 | 58.41% ± 2.18 | 68.47% ± 1.86 | 69.13% ± 1.71 | 73.91% ± 1.67 | 75.53% ± 1.58 | 76.85% ± 1.55 |
| GPT-4o Mini | 58.81% ± 2.25 | 68.85% ± 2.07 | 71.60% ± 1.84 | 74.71% ± 1.83 | 77.66% ± 1.70 | 79.52% ± 1.53 |
| GPT-4o | 58.61% ± 2.30 | 67.57% ± 2.18 | 71.37% ± 1.85 | 75.03% ± 1.83 | 77.62% ± 1.70 | 79.46% ± 1.60 |

| | Qwen 72B | Llama 8B | Llama 70B | DeepSeekV3 | GPT-4o Mini | GPT-4o |
|---|---|---|---|---|---|---|
| Qwen 0.5B | 60.79% ± 2.38 | 62.77% ± 2.13 | 60.61% ± 2.37 | 59.51% ± 2.44 | 60.85% ± 2.32 | 58.50% ± 2.48 |
| Qwen 1.5B | 74.23% ± 1.81 | 72.89% ± 1.69 | 73.18% ± 1.85 | 73.22% ± 1.94 | 71.95% ± 1.89 | 71.72% ± 2.04 |
| Qwen 3B | 71.15% ± 1.77 | 71.08% ± 1.69 | 70.79% ± 1.79 | 70.66% ± 1.83 | 71.15% ± 1.70 | 70.03% ± 1.82 |
| Qwen 7B | 73.62% ± 1.66 | 71.53% ± 1.69 | 73.36% ± 1.67 | 73.32% ± 1.75 | 71.54% ± 1.75 | 72.46% ± 1.80 |
| Qwen 14B | 73.32% ± 1.64 | 70.07% ± 1.71 | 72.94% ± 1.69 | 72.48% ± 1.75 | 72.09% ± 1.73 | 72.53% ± 1.78 |
| Qwen 32B | 78.74% ± 1.46 | 72.96% ± 1.69 | 78.28% ± 1.50 | 77.95% ± 1.59 | 75.94% ± 1.65 | 77.78% ± 1.64 |
| Qwen 72B | **80.22%** ± **1.36** | 72.10% ± 1.71 | 77.50% ± 1.53 | 78.20% ± 1.51 | 74.52% ± 1.73 | 76.43% ± 1.65 |
| Llama 8B | 75.64% ± 1.71 | **80.42%** ± **1.30** | 76.43% ± 1.65 | 75.65% ± 1.74 | 74.70% ± 1.75 | 74.73% ± 1.85 |
| Llama 70B | 79.82% ± 1.49 | 74.16% ± 1.70 | **83.22%** ± **1.29** | 79.27% ± 1.60 | 78.07% ± 1.62 | 79.35% ± 1.61 |
| DeepSeekV3 | 78.67% ± 1.49 | 72.02% ± 1.70 | 77.18% ± 1.63 | **81.20%** ± **1.44** | 74.70% ± 1.76 | 77.06% ± 1.71 |
| GPT-4o Mini | 80.95% ± 1.59 | 74.53% ± 1.84 | 80.55% ± 1.64 | 81.44% ± 1.63 | **82.88%** ± **1.50** | 81.75% ± 1.56 |
| GPT-4o | 81.79% ± 1.54 | 73.65% ± 1.92 | 80.76% ± 1.69 | 82.81% ± 1.51 | 79.56% ± 1.67 | **84.90%** ± **1.39** |

Table 26: Accuracy of proxy Kernel SHAP explanations on high school psychology of MMLU datasets: each value shows how well Kernel SHAP explanations generated by the model on the **left** serve as surrogates for predicting the behavior of the model on the **top**.

|  | Qwen 0.5B | Qwen 1.5B | Qwen 3B | Qwen 7B | Qwen 14B | Qwen 32B |
|---|---|---|---|---|---|---|
| Qwen 0.5B | **79.08%** ± **1.31** | 74.22% ± 1.95 | 72.93% ± 1.87 | 74.33% ± 1.83 | 74.51% ± 1.99 | 74.74% ± 1.92 |
| Qwen 1.5B | 71.53% ± 2.08 | **79.61%** ± **1.33** | 75.67% ± 1.54 | 77.69% ± 1.46 | 77.34% ± 1.58 | 77.96% ± 1.53 |
| Qwen 3B | 67.79% ± 2.10 | 72.71% ± 1.68 | **75.48%** ± **1.39** | 73.61% ± 1.56 | 74.67% ± 1.54 | 74.65% ± 1.56 |
| Qwen 7B | 68.82% ± 2.03 | 72.49% ± 1.63 | 71.43% ± 1.60 | **77.64%** ± **1.38** | 75.88% ± 1.46 | 76.06% ± 1.47 |
| Qwen 14B | 68.01% ± 2.08 | 71.32% ± 1.68 | 71.58% ± 1.61 | 73.57% ± 1.55 | **76.59%** ± **1.47** | 75.41% ± 1.56 |
| Qwen 32B | 68.95% ± 2.20 | 72.59% ± 1.82 | 73.47% ± 1.66 | 76.69% ± 1.58 | 78.71% ± 1.49 | **80.23%** ± **1.32** |
| Qwen 72B | 67.71% ± 2.22 | 72.93% ± 1.74 | 73.09% ± 1.60 | 76.02% ± 1.47 | 77.49% ± 1.48 | 77.89% ± 1.38 |
| Llama 8B | 68.90% ± 2.22 | 74.14% ± 1.80 | 74.68% ± 1.63 | 77.49% ± 1.56 | 78.30% ± 1.53 | 79.23% ± 1.39 |
| Llama 70B | 69.71% ± 2.34 | 73.84% ± 1.85 | 73.70% ± 1.73 | 77.59% ± 1.55 | 79.26% ± 1.50 | 79.94% ± 1.40 |
| DeepSeekV3 | 66.84% ± 2.25 | 72.66% ± 1.82 | 72.10% ± 1.72 | 75.88% ± 1.60 | 77.05% ± 1.53 | 78.14% ± 1.47 |
| GPT-4o Mini | 68.49% ± 2.40 | 74.08% ± 1.93 | 74.62% ± 1.81 | 77.38% ± 1.70 | 79.86% ± 1.59 | 81.34% ± 1.41 |
| GPT-4o | 67.44% ± 2.51 | 73.06% ± 2.08 | 74.54% ± 1.85 | 77.72% ± 1.72 | 79.61% ± 1.65 | 81.15% ± 1.47 |

|  | Qwen 72B | Llama 8B | Llama 70B | DeepSeekV3 | GPT-4o Mini | GPT-4o |
|---|---|---|---|---|---|---|
| Qwen 0.5B | 75.51% ± 1.92 | 73.90% ± 1.98 | 75.32% ± 1.96 | 74.36% ± 2.01 | 74.99% ± 1.93 | 74.59% ± 2.06 |
| Qwen 1.5B | 79.37% ± 1.53 | 76.60% ± 1.60 | 78.11% ± 1.60 | 79.03% ± 1.62 | 77.55% ± 1.63 | 78.58% ± 1.61 |
| Qwen 3B | 74.62% ± 1.61 | 74.37% ± 1.60 | 74.32% ± 1.62 | 74.70% ± 1.67 | 74.17% ± 1.57 | 74.33% ± 1.64 |
| Qwen 7B | 75.52% ± 1.54 | 74.00% ± 1.63 | 75.52% ± 1.57 | 75.61% ± 1.61 | 74.25% ± 1.56 | 75.82% ± 1.59 |
| Qwen 14B | 74.74% ± 1.64 | 72.66% ± 1.69 | 74.33% ± 1.67 | 74.16% ± 1.68 | 74.20% ± 1.60 | 75.02% ± 1.65 |
| Qwen 32B | 79.76% ± 1.41 | 75.96% ± 1.63 | 79.07% ± 1.51 | 79.39% ± 1.48 | 77.99% ± 1.49 | 79.70% ± 1.47 |
| Qwen 72B | **80.22%** ± **1.36** | 74.88% ± 1.68 | 78.28% ± 1.51 | 78.84% ± 1.49 | 76.69% ± 1.52 | 77.78% ± 1.53 |
| Llama 8B | 79.35% ± 1.49 | **80.42%** ± **1.30** | 79.59% ± 1.49 | 79.22% ± 1.49 | 78.34% ± 1.49 | 79.08% ± 1.55 |
| Llama 70B | 81.02% ± 1.40 | 77.30% ± 1.64 | **83.22%** ± **1.29** | 80.92% ± 1.48 | 80.10% ± 1.41 | 81.31% ± 1.41 |
| DeepSeekV3 | 79.29% ± 1.49 | 74.58% ± 1.67 | 78.34% ± 1.58 | **81.20%** ± **1.44** | 76.84% ± 1.57 | 78.42% ± 1.58 |
| GPT-4o Mini | 82.86% ± 1.43 | 77.76% ± 1.77 | 82.32% ± 1.49 | 83.53% ± 1.43 | **82.88%** ± **1.50** | 83.21% ± 1.43 |
| GPT-4o | 83.03% ± 1.42 | 77.11% ± 1.84 | 82.22% ± 1.56 | 83.76% ± 1.45 | 81.02% ± 1.57 | **84.90%** ± **1.39** |

Table 27: Accuracy of **filtered** proxy Kernel SHAP explanations on high school psychology of MMLU datasets: each value shows how well Kernel SHAP explanations generated by the model on the **left** serve as surrogates for predicting the behavior of the model on the **top**.

|  | Qwen 0.5B | Qwen 1.5B | Qwen 3B | Qwen 7B | Qwen 14B | Qwen 32B |
|---|---|---|---|---|---|---|
| Qwen 0.5B | **72.40%** ± **2.05** | 62.35% ± 2.94 | 59.56% ± 3.24 | 60.02% ± 3.37 | 59.10% ± 3.42 | 57.78% ± 3.57 |
| Qwen 1.5B | 60.54% ± 2.90 | **71.46%** ± **2.21** | 66.05% ± 2.75 | 64.90% ± 2.85 | 65.14% ± 2.98 | 64.04% ± 3.11 |
| Qwen 3B | 57.08% ± 3.00 | 63.65% ± 2.78 | **69.47%** ± **2.53** | 66.65% ± 2.70 | 67.13% ± 2.85 | 65.96% ± 3.00 |
| Qwen 7B | 57.38% ± 3.22 | 63.31% ± 2.95 | 65.71% ± 2.78 | **69.57%** ± **2.58** | 68.18% ± 2.77 | 67.32% ± 2.88 |
| Qwen 14B | 57.35% ± 3.19 | 61.86% ± 3.11 | 65.27% ± 2.96 | 67.56% ± 2.87 | **70.78%** ± **2.75** | 69.59% ± 2.89 |
| Qwen 32B | 59.10% ± 3.23 | 63.17% ± 3.17 | 66.26% ± 3.02 | 70.14% ± 2.82 | 73.02% ± 2.72 | **73.87%** ± **2.68** |
| Qwen 72B | 59.32% ± 3.36 | 63.77% ± 3.26 | 68.47% ± 3.07 | 70.72% ± 2.87 | 73.89% ± 2.72 | 74.45% ± 2.81 |
| Llama 8B | 59.53% ± 3.05 | 64.20% ± 2.79 | 68.15% ± 2.66 | 69.40% ± 2.56 | 71.08% ± 2.51 | 71.23% ± 2.54 |
| Llama 70B | 60.00% ± 3.40 | 63.85% ± 3.35 | 68.61% ± 3.08 | 71.83% ± 2.83 | 75.21% ± 2.66 | 75.08% ± 2.75 |
| DeepSeekV3 | 59.74% ± 3.57 | 65.58% ± 3.40 | 68.28% ± 3.28 | 71.61% ± 2.97 | 74.67% ± 2.88 | 74.50% ± 2.94 |
| GPT-4o Mini | 59.78% ± 3.31 | 64.68% ± 3.17 | 67.96% ± 2.93 | 71.22% ± 2.80 | 73.81% ± 2.66 | 73.81% ± 2.78 |
| GPT-4o | 59.88% ± 3.45 | 64.76% ± 3.37 | 68.83% ± 3.18 | 73.11% ± 2.86 | 76.28% ± 2.67 | 76.22% ± 2.76 |

|  | Qwen 72B | Llama 8B | Llama 70B | DeepSeekV3 | GPT-4o Mini | GPT-4o |
|---|---|---|---|---|---|---|
| Qwen 0.5B | 57.37% ± 3.66 | 57.92% ± 3.39 | 56.94% ± 3.60 | 57.64% ± 3.71 | 58.07% ± 3.58 | 56.82% ± 3.78 |
| Qwen 1.5B | 63.60% ± 3.31 | 63.93% ± 3.00 | 62.70% ± 3.19 | 64.44% ± 3.31 | 63.97% ± 3.20 | 62.38% ± 3.43 |
| Qwen 3B | 65.84% ± 3.11 | 65.85% ± 2.83 | 64.87% ± 3.09 | 65.27% ± 3.18 | 65.55% ± 2.95 | 65.44% ± 3.20 |
| Qwen 7B | 67.86% ± 2.96 | 66.24% ± 2.85 | 66.17% ± 2.97 | 67.30% ± 3.02 | 66.50% ± 3.01 | 67.81% ± 2.96 |
| Qwen 14B | 69.76% ± 3.00 | 65.89% ± 2.95 | 68.90% ± 2.97 | 69.73% ± 2.99 | 68.22% ± 2.96 | 70.00% ± 3.04 |
| Qwen 32B | 72.97% ± 2.87 | 68.77% ± 2.94 | 71.51% ± 2.92 | 72.60% ± 2.90 | 71.18% ± 2.87 | 72.93% ± 2.94 |
| Qwen 72B | **76.75%** ± **2.69** | 69.72% ± 3.06 | 74.91% ± 2.80 | 74.89% ± 2.87 | 73.34% ± 2.86 | 75.99% ± 2.85 |
| Llama 8B | 71.62% ± 2.74 | **72.90%** ± **2.34** | 70.23% ± 2.76 | 70.89% ± 2.81 | 69.83% ± 2.76 | 70.73% ± 2.87 |
| Llama 70B | 76.64% ± 2.69 | 71.32% ± 2.98 | **78.17%** ± **2.49** | 77.25% ± 2.66 | 74.90% ± 2.78 | 78.10% ± 2.60 |
| DeepSeekV3 | 76.17% ± 2.91 | 70.64% ± 3.15 | 75.11% ± 2.91 | **78.56%** ± **2.75** | 74.02% ± 2.95 | 77.89% ± 2.82 |
| GPT-4o Mini | 75.06% ± 2.73 | 70.99% ± 2.82 | 74.50% ± 2.74 | 75.02% ± 2.74 | **75.64%** ± **2.61** | 75.44% ± 2.78 |
| GPT-4o | 78.32% ± 2.70 | 72.02% ± 3.01 | 76.65% ± 2.78 | 78.78% ± 2.66 | 75.69% ± 2.80 | **80.01%** ± **2.60** |

Table 28: Accuracy of proxy Kernel SHAP explanations on high school world history of MMLU datasets: each value shows how well Kernel SHAP explanations generated by the model on the **left** serve as surrogates for predicting the behavior of the model on the **top**.

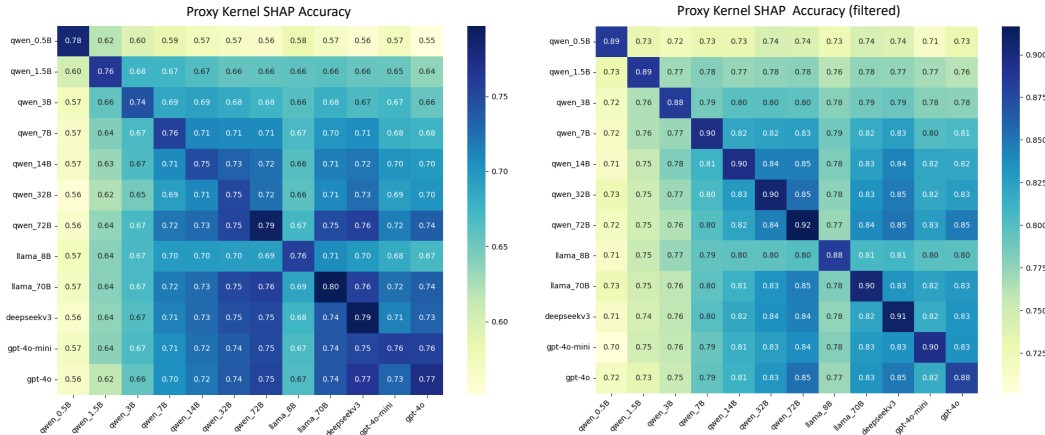

Figure 5: Accuracy of Kernel SHAP proxy explanations on the multiple-choice question answering task. Each cell shows how well explanations generated by the model on the **y-axis** serve as surrogates for predicting the behavior of the model on the **x-axis**. The heatmap on the right shows results after filtering out examples where the budget-friendly and expensive models produce different predictions for the input.

|  | Qwen 0.5B | Qwen 1.5B | Qwen 3B | Qwen 7B | Qwen 14B | Qwen 32B |
|---|---|---|---|---|---|---|
| Qwen 0.5B | **72.40% ± 2.05** | 69.14% ± 3.13 | 67.89% ± 3.27 | 68.97% ± 3.25 | 70.98% ± 3.15 | 71.14% ± 3.17 |
| Qwen 1.5B | 66.58% ± 3.17 | **71.46% ± 2.21** | 68.56% ± 2.84 | 69.61% ± 2.97 | 70.99% ± 2.91 | 70.67% ± 2.88 |
| Qwen 3B | 63.13% ± 3.33 | 67.12% ± 2.93 | **69.47% ± 2.53** | 67.74% ± 2.89 | 69.78% ± 2.99 | 69.41% ± 3.01 |
| Qwen 7B | 65.07% ± 3.33 | 68.07% ± 2.98 | 66.88% ± 2.91 | **69.57% ± 2.58** | 69.82% ± 2.81 | 69.69% ± 2.87 |
| Qwen 14B | 65.76% ± 3.52 | 67.57% ± 3.23 | 67.93% ± 3.12 | 69.01% ± 3.01 | **70.78% ± 2.75** | 71.04% ± 2.96 |
| Qwen 32B | 67.16% ± 3.43 | 68.75% ± 3.20 | 69.01% ± 3.13 | 71.31% ± 2.97 | 73.79% ± 2.82 | **73.87% ± 2.68** |
| Qwen 72B | 68.43% ± 3.40 | 70.73% ± 3.18 | 71.74% ± 3.07 | 72.86% ± 2.93 | 76.04% ± 2.73 | 76.19% ± 2.85 |
| Llama 8B | 67.43% ± 3.08 | 68.96% ± 2.83 | 69.79% ± 2.73 | 71.56% ± 2.66 | 73.38% ± 2.60 | 73.31% ± 2.57 |
| Llama 70B | 69.79% ± 3.46 | 71.62% ± 3.15 | 72.45% ± 3.07 | 73.81% ± 2.87 | 77.56% ± 2.65 | 77.43% ± 2.67 |
| DeepSeekV3 | 69.40% ± 3.60 | 73.27% ± 3.20 | 71.65% ± 3.33 | 73.29% ± 3.05 | 76.57% ± 2.92 | 75.87% ± 2.98 |
| GPT-4o Mini | 68.42% ± 3.34 | 71.10% ± 3.19 | 70.72% ± 2.99 | 73.15% ± 2.84 | 75.70% ± 2.73 | 75.83% ± 2.79 |
| GPT-4o | 70.26% ± 3.42 | 71.91% ± 3.23 | 72.42% ± 3.15 | 74.68% ± 2.93 | 78.09% ± 2.71 | 77.61% ± 2.76 |

|  | Qwen 72B | Llama 8B | Llama 70B | DeepSeekV3 | GPT-4o Mini | GPT-4o |
|---|---|---|---|---|---|---|
| Qwen 0.5B | 70.14% ± 3.28 | 68.51% ± 3.28 | 70.02% ± 3.38 | 71.42% ± 3.29 | 69.45% ± 3.32 | 71.82% ± 3.36 |
| Qwen 1.5B | 71.28% ± 2.97 | 69.62% ± 2.99 | 70.66% ± 2.98 | 72.63% ± 2.97 | 70.39% ± 3.11 | 71.47% ± 3.02 |
| Qwen 3B | 69.69% ± 3.05 | 67.93% ± 2.90 | 68.83% ± 3.07 | 69.01% ± 3.11 | 68.67% ± 2.98 | 69.69% ± 3.10 |
| Qwen 7B | 70.35% ± 2.95 | 68.64% ± 2.94 | 68.63% ± 2.96 | 69.16% ± 3.01 | 68.93% ± 3.02 | 70.27% ± 2.92 |
| Qwen 14B | 71.77% ± 3.03 | 68.51% ± 3.06 | 71.09% ± 3.05 | 71.56% ± 3.02 | 70.13% ± 3.06 | 72.33% ± 3.03 |
| Qwen 32B | 74.40% ± 2.93 | 70.83% ± 3.00 | 73.21% ± 2.98 | 73.51% ± 2.97 | 72.84% ± 2.95 | 74.54% ± 2.89 |
| Qwen 72B | **76.75% ± 2.69** | 72.92% ± 3.05 | 75.98% ± 2.85 | 76.64% ± 2.84 | 74.83% ± 2.86 | 77.75% ± 2.81 |
| Llama 8B | 74.93% ± 2.61 | **72.90% ± 2.34** | 74.10% ± 2.69 | 74.53% ± 2.71 | 73.25% ± 2.72 | 74.71% ± 2.64 |
| Llama 70B | 77.93% ± 2.69 | 75.18% ± 2.89 | **78.17% ± 2.49** | 78.06% ± 2.63 | 76.92% ± 2.70 | 79.13% ± 2.56 |
| DeepSeekV3 | 77.54% ± 2.96 | 74.17% ± 3.14 | 75.72% ± 2.97 | **78.56% ± 2.75** | 75.43% ± 2.95 | 78.69% ± 2.85 |
| GPT-4o Mini | 76.36% ± 2.73 | 73.80% ± 2.85 | 75.96% ± 2.80 | 76.53% ± 2.69 | **75.64% ± 2.61** | 77.85% ± 2.73 |
| GPT-4o | 79.76% ± 2.66 | 75.24% ± 2.92 | 77.41% ± 2.81 | 79.46% ± 2.69 | 77.55% ± 2.82 | **80.01% ± 2.60** |

Table 29: Accuracy of **filtered** proxy Kernel SHAP explanations on high school world history of MMLU datasets: each value shows how well Kernel SHAP explanations generated by the model on the **left** serve as surrogates for predicting the behavior of the model on the **top**.

provide the detailed results with 95% confidence intervals in Table table 10, 11, 12, 13, 14, 15, 16, 17, 18, 19, 20, 21, 22, 23, 24, 25, 26, 27, 28, and 29.

Another notable observation is the fidelity of oracle explanations also differ in different subjects. For `high school microeconomics`, `high school psychology`, and `high school world history`, the oracle LIME explanations generated by the model all achieve a fidelity higher than 90%, while for `high school computer science`, `high school chemistry`, and `high school physics`, the fidelity is relative lower. The subjects with higher fidelity are all related to social sciences, while the subjects with lower fidelity are all related

to natural sciences. This may be due to the fact that social science questions often have more diverse and complex answer options, leading performace differences between models.

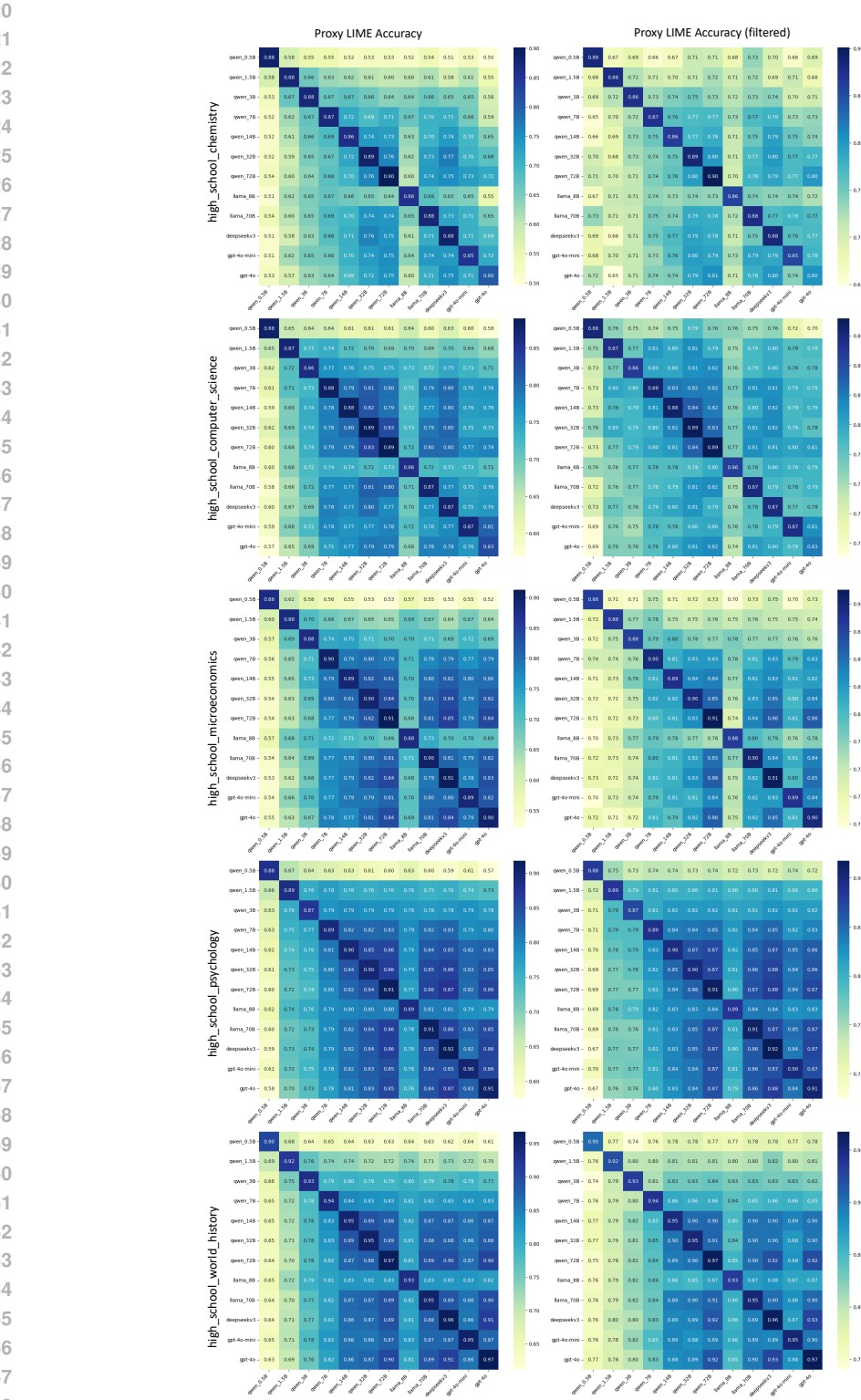

Figure 6: Accuracy of LIME proxy explanations on the multiple-choice question answering task on each subject. Each cell shows how well explanations generated by the model on the **y-axis** serve as surrogates for predicting the behavior of the model on the **x-axis**. The heatmap on the right shows results after filtering out examples where the budget-friendly and expensive models produce different predictions for the input.

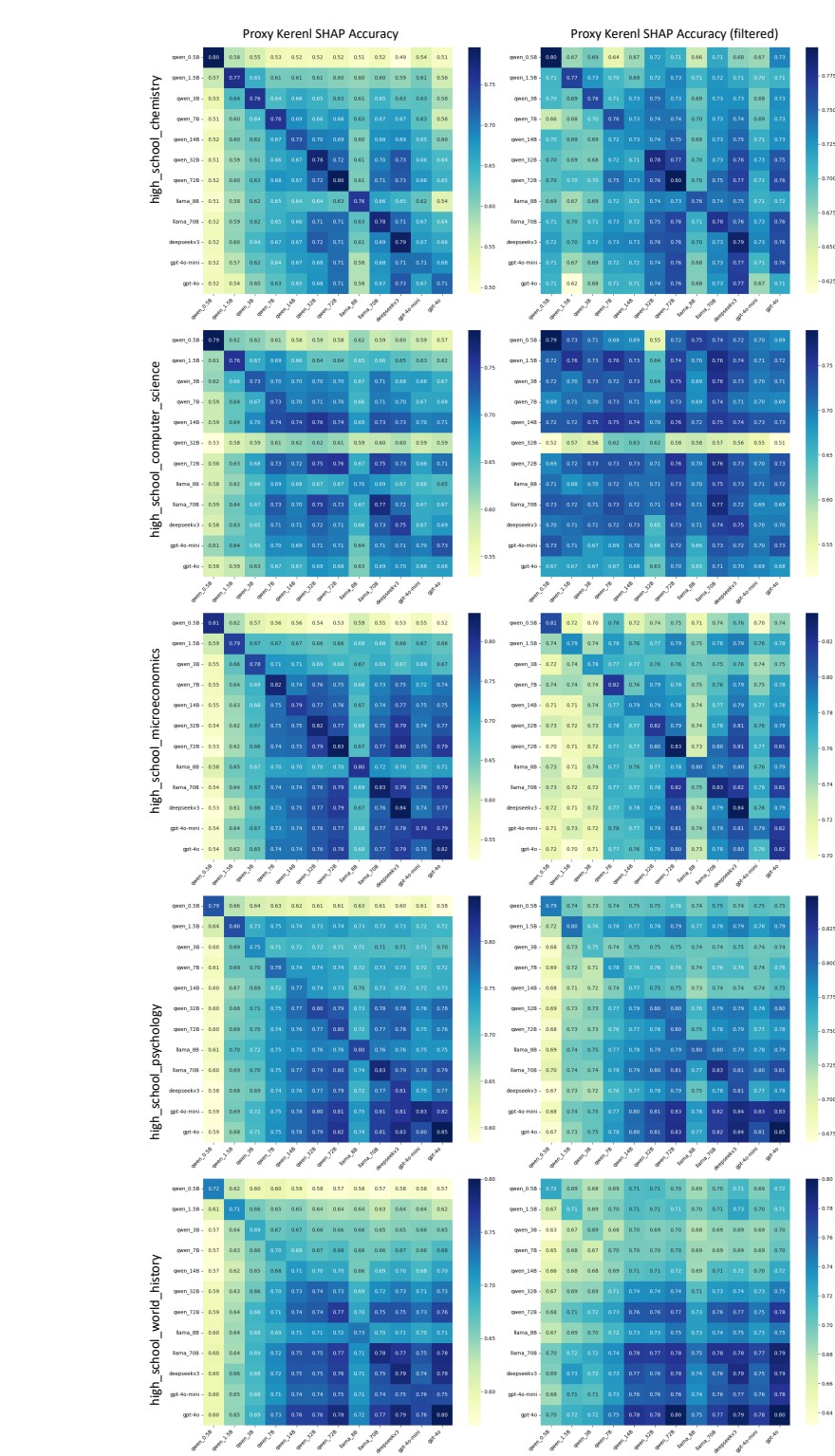

Figure 7: Accuracy of Kernel SHAP proxy explanations on the multiple-choice question answering task on each subject. Each cell shows how well explanations generated by the model on the **y-axis** serve as surrogates for predicting the behavior of the model on the **x-axis**. The heatmap on the right shows results after filtering out examples where the budget-friendly and expensive models produce different predictions for the input.

