# OpenReview forum: "See the Big in the Small: Budget-Friendly Explanations for Large Language Models"
_ICLR.cc/2026/Conference — ICLR 2026 Conference Withdrawn Submission_

### Official Review · Reviewer_vDdu · 2025-10-30

**Soundness:** 2
**Presentation:** 3
**Contribution:** 2
**Rating:** 4
**Confidence:** 3

**Summary:**

This paper aims to reduce the cost associated with generating local explanations for LLMs. The authors propose a novel screen-and-apply framework that leverages budget-friendly (e.g., smaller, open-source) models to generate faithful "proxy explanations" for more expensive target LLMs. The core of the framework is a two-stage screening process designed to ensure the reliability of the proxy explanations.

**Strengths:**

- The two-stage screening process is methodologically sound. The task-level screen is formally defined as a sequential paired t-test to provide statistical confidence that a proxy model is suitable on average. The instance-level screen is an intuitive filter based on the key insight that explanations are unlikely to transfer if the models' primary predictions disagree.
- The empirical evaluation is thorough with diverse LLMs, different tasks across multiple domains (classification, QA, generation), and two widely-used explanation techniques (LIME, SHAP).
- The paper is well-written. The motivation is established clearly in the introduction with a concrete, compelling cost example. The proposed framework is described logically and is easy to understand. The figures and tables are clear and support the paper's claims.

**Weaknesses:**

- A significant weakness is the lack of detail regarding the core explanation mechanisms. The authors do not describe the explanation units (e.g., tokens, words, n-grams) that serve as features for LIME and SHAP. Furthermore, the paper omits how these features are modified or perturbed to generate local samples. This is a critical detail for reproducibility and understanding the nature of the explanations. The fidelity metric is defined as $\mathbb{E}_{\\mathbf{z} \sim D(\\mathbf{x})}L(f(\\mathbf{z}),g(\\mathbf{z}))$, but the neighborhood distribution $D(\\mathbf{x})$ is never specified, making it impossible to fully interpret the fidelity scores.
- The paper focuses almost exclusively on cases where high-fidelity proxy explanations are successfully found. It does not adequately discuss or analyze scenarios where the screening process fails and no suitable proxy model passes the task-level screen. This focus on positive results misrepresents the practical applicability of the framework and fails to explore its limitations. A robust analysis would investigate why certain models fail the screen and what a user's recourse is when no budget-friendly model is deemed faithful. This would provide a more balanced view of the technique's boundaries.
- The paper does not investigate whether the faithfulness of a proxy-target model pair is transferable across different tasks. The current framework assumes that screening must be done on a per-task basis, but it misses a crucial analysis: Is the screening performance of a proxy model on one task indicative of its performance on another? The empirical results suggest the answer is no, but the authors fail to explore this.
- The numerical evaluation of the screening process in Section 4.3.2 focuses exclusively on the reliability of the task-level screen. However, the contribution of the instance-level screen is only demonstrated visually through the "filtered" heatmaps in the appendix. A direct, quantitative evaluation of this critical step is missing.
- The paper presents its extensive empirical results almost exclusively through large, dense tables, particularly in the appendix. While these tables are valuable for completeness, their format makes it difficult for the reader to quickly grasp qualitative patterns. For example, questions like "Is there a clear relationship between proxy model size and its fidelity?" or "Do models from the same family consistently serve as better proxies for each other?" are hard to answer by scanning dozens of numerical entries. The inclusion of more visualizations (scatter plots, bar plots, etc.) would have made the key takeaways much more immediate and intuitive.

**Questions:**

- The central motivation for this work is the observation that "when [different models] produce the same outputs, they also tend to behave alike on similar inputs". Beyond the success of the overall framework, what specific analysis was done to directly verify this foundational claim?
- The overall cost reduction depends heavily on the prediction agreement rate between the proxy and target models (i.e., the success rate of the instance-level screen). Could you provide a brief breakdown of this agreement rate for the different proxy-target pairs and tasks in your experiments?
- In Section 4.3.2, you evaluate the screening step using precision, recall, and F1-score. Could you please clarify what constitutes the ground truth label for this binary classification task? Specifically, how do you determine if a screening decision (pass/fail) is a true positive, false positive, true negative, or false negative?
- The results primarily focus on proxy-target pairs that successfully pass the task-level screening. Were there any instances where a candidate proxy model failed the task-level screening (i.e., was deemed not sufficiently faithful)? Presenting and analyzing these failure cases would be highly valuable for understanding the limitations and boundaries of the proposed technique.
- After a proxy has passed the task-level screen, what is the specific, quantitative improvement in fidelity gained by applying the additional instance-level filter?
- Beyond the current implementation, have the authors considered enhancements to the instance-level screening process? For example, could metrics other than simple prediction agreement, such as the confidence of the models' predictions or the entropy of the output distributions, be used to create a more nuanced and effective filter for determining when to use the proxy model?

---

### Official Review · Reviewer_do7q · 2025-11-01

**Soundness:** 2
**Presentation:** 2
**Contribution:** 2
**Rating:** 2
**Confidence:** 4

**Summary:**

The paper proposes a proxy explanation framework for large language models (LLMs), aiming to reduce the cost of generating local explanations by using smaller “budget-friendly” models as substitutes. The framework introduces a two-stage screening process, task-level statistical testing and instance-level agreement filtering, to ensure the fidelity of proxy explanations. Experiments on several LLMs across multiple datasets evaluate cost reduction, screening reliability, and explanation generalizability. The authors report that their method can reduce explanation cost by up to 88.2% while maintaining comparable fidelity to oracle explanations.

**Strengths:**

1. The paper addresses an important and practical problem: black-box, model-agnostic local explanations typically require extensive querying, which becomes prohibitively expensive when using commercial LLM APIs. The authors propose a pragmatic compromise to reduce explanation cost.
2. The idea of using smaller “budget-friendly” models as proxies for explanation is conceptually interesting, and the proposed framework is simple and easy to implement. The instance-level screening only requires label agreement between models, making it practical in engineering settings, while the task-level screening based on sequential paired t-tests is intuitive and reproducible.
3. The experiments are broad and well-organized, covering multiple models and tasks.

**Weaknesses:**

1. The core assumption lacks sufficient validation. The paper’s central assumption is that when two models produce the same output, their local decision behaviors are also similar, allowing explanations to be transferred. However: The screening mechanism relies solely on output-level label agreement f(x)=f′(x). The paper provides no theoretical analysis or mechanistic investigation explaining why output agreement would imply similarity at the reasoning or explanation level. Two models may reach the same output through entirely different reasoning paths, for example, a large model might rely on deep semantic reasoning, whereas a smaller model might use surface-level pattern matching, but the paper does not investigate this possibility.
2. The task-level and instance-level screening criteria partially overlap. Both stages rely on output agreement, which limits the conceptual novelty of the proposed “two-stage” design. The paper claims that the task-level screening verifies overall applicability while the instance-level screening filters individual mismatched cases. In practice, however, the two stages do not provide complementary safeguards; they merely apply the same condition, output agreement, at different granularities (aggregate statistics vs. per-instance checks).
3. The cost–benefit analysis has some practical limitations. Although the CRR formula in the paper includes the cost of a single screening run, several key factors in practical deployment are insufficiently discussed. First, finding the optimal proxy model requires testing multiple candidates, which multiplies the screening cost, whereas this paper assumes the best proxy is already known. Second, all experiments are conducted on large-scale datasets; for small-batch queries, the fixed screening overhead could exceed the cost of directly using the oracle, yet the paper does not analyze this trade-off point. Finally, screening requires pre-collected samples, making the method unsuitable for cold-start or one-off, real-time queries. Therefore, the reported 88.2% cost reduction reflects an idealized scenario (known optimal proxy, large-batch processing, and offline setup), while the actual savings in realistic settings are likely to be much lower.

**Questions:**

1. Could the authors provide evidence that output-level agreement also indicates similar local decision behavior?
2. Given that task-level screening is performed only on samples where f(x) = f'(x), and instance-level screening checks the same condition, how do the two stages provide complementary safeguards beyond applying the same criterion at different granularities?

---

### Official Review · Reviewer_xfoC · 2025-11-03

**Soundness:** 2
**Presentation:** 3
**Contribution:** 3
**Rating:** 4
**Confidence:** 4

**Summary:**

This paper addresses a key challenge in the interpretability of large language models (LLMs): the high cost of generating explanations, since classical interpretability methods often require numerous or even exponential queries to closed-source models (e.g., GPT-4o). To mitigate this issue, the authors propose a surrogate explanation framework, which employs a low-cost small model to generate substitute explanations for expensive large models. To ensure reliability and consistency, a “screen-and-apply” dual filtering mechanism is introduced, combining task-level sequential t-tests with instance-level consistency checks. Experiments conducted across multiple LLMs, tasks, and datasets demonstrate that the proposed method reduces cost by up to 88.2% while maintaining over 90% fidelity. In addition, the authors release XLLM-Bench, a dataset containing over 3.7 million perturbed samples with corresponding model outputs, providing a valuable resource for the community.

**Strengths:**

1. It directly focus on the economic bottleneck of LLM interpretability (e.g., a single explanation on GPT-4o costs around $12.5).

2. It is interesting that small model can approximately interpret large models.

3. The release of the large-scale XLLM-Bench dataset enhances the openness and reproducibility of the LLM interpretability research.

**Weaknesses:**

1. Lack of theoretical foundation: The paper does not explain why or under what conditions a small model can reliably approximate the interpretive behavior of a large model, leaving the applicability boundaries of the method unclear.

2. The assessment of explanation fidelity relies on a few metrics, which may not comprehensively capture the quality of surrogate explanations. Introducing more metrics could improve objectivity and robustness.

3. The paper does not discuss which types of small models are suitable for interpreting large models, for example what characteristics (e.g., architectural similarity, capacity alignment, training data overlap) they should possess.

4. Insufficient generalization validation: The experiments cover a limited number of datasets (mostly three), making it difficult to determine whether surrogate explanations remain consistent and reliable across diverse tasks and data distributions.

5. When the most effective surrogate models have performance or parameter scales close to the original model, the cost advantage diminishes, reducing the method’s practicality and scalability.

**Questions:**

Please see the weakness.

---

### Official Review · Reviewer_Joya · 2025-11-03

**Soundness:** 3
**Presentation:** 3
**Contribution:** 1
**Rating:** 2
**Confidence:** 4

**Summary:**

This paper proposes a proxy-based framework for generating cost-efficient local explanations of LLMs. Instead of querying expensive models like GPT-4o directly with perturbation-based methods (e.g., LIME, SHAP), the authors suggest using explanations from smaller, “budget-friendly” models as surrogates. A two-stage screen-and-apply process first tests whether a proxy’s explanations are sufficiently faithful through statistical screening, then applies them if the models’ predictions align. Experiments on twelve LLMs across several NLP tasks show over 90% fidelity compared to oracle explanations, with an 88% reduction in cost.

**Strengths:**

The paper is technically well-executed and introduces a systematic way to formalize “explanation transfer” between models through a screening process based on sequential one-sided t-tests. The formulation of fidelity as a hypothesis-testing criterion is a clean, rigorous addition to the literature on model-agnostic interpretability. The authors evaluate across diverse models and tasks, demonstrating practical relevance for real-world API-based LLM scenarios. The inclusion of XLLM-Bench also contributes useful benchmarking data for further research.

**Weaknesses:**

- The paper’s main contribution is incremental relative to prior efforts in efficient post-hoc explainability. The proposed screen-and-apply framework builds primarily on existing components: model-agnostic explainers, local neighborhood sampling, and standard significance testing, rather than introducing new algorithmic mechanisms or theoretical insights about why proxy models’ explanations should transfer across scales.

- The idea that smaller models can approximate larger ones when outputs agree is intuitive but weakly justified beyond empirical correlation. The screening’s reliance on sequential t-tests assumes independence of perturbation samples, which is unrealistic under correlated neighborhood sampling.

- Moreover, the hyperparameters (τ, δ, N) are chosen heuristically, and the paper does not analyze the statistical power or false acceptance rate of the screening procedure.

- The fidelity metric used (local surrogate accuracy) measures consistency but not interpretive faithfulness or causal alignment, meaning proxy explanations may match labels while misrepresenting reasoning.

- Cross-model agreement is used as a proxy for behavioral similarity, yet this is insufficient for complex reasoning tasks or for models differing in training objectives.

- The case studies on prompt compression and poisoned example removal are superficial demonstrations rather than validations of interpretability quality. In addition, comparisons are missing against amortized or learned explanation methods (e.g., FastSHAP, CortX, Covert et al. 2024), which already address efficiency concerns more directly.

**Questions:**

1. How robust are the screening decisions to correlated perturbations and non-Gaussian fidelity distributions?

2. Can the authors theoretically bound the fidelity gap introduced by proxy substitution?

3. Would amortized explanation models or meta-explainers trained across LLMs yield similar or better efficiency without screening overhead?

4. How sensitive are results to τ and δ, and can adaptive calibration mitigate false acceptance?

5. Do the authors have evidence that explanations remain causally faithful when the proxy and target differ substantially in architecture (e.g., MoE vs. dense)?

**Details Of Ethics Concerns:**

None.

---

### Note · Authors · 2026-01-06

I have read and agree with the venue's withdrawal policy on behalf of myself and my co-authors.